# A Critical Review of the Current Global Ex Situ Conservation System for Plant Agrobiodiversity. I. History of the Development of the Global System in the Context of the Political/Legal Framework and Its Major Conservation Components

**DOI:** 10.3390/plants10081557

**Published:** 2021-07-29

**Authors:** Johannes M. M. Engels, Andreas W. Ebert

**Affiliations:** 1Alliance of Bioversity International and CIAT, 00153 Rome, Italy; 2World Vegetable Center, 60 Yi-Min Liao, Shanhua, Tainan 74151, Taiwan; ebert.andreas6@gmail.com

**Keywords:** plant agrobiodiversity, history of the global ex situ conservation system, political and legal framework, field genebanks, in vitro collections, cryopreservation, DNA banks, pollen banks, complementary conservation approaches

## Abstract

The history of ex situ conservation is relatively short, not more than a century old. During the middle of last century, triggered by the realization that genetic erosion was threatening the existing landraces and wild relatives of the major food crops, global efforts to collect and conserve the genetic diversity of these threatened resources were initiated, predominantly orchestrated by FAO. National and international genebanks were established to store and maintain germplasm materials, conservation methodologies were created, standards developed, and coordinating efforts were put in place to ensure effective and efficient approaches and collaboration. In the spontaneously developing global conservation system, plant breeders played an important role, aiming at the availability of genetic diversity in their breeding work. Furthermore, long-term conservation and the safety of the collected materials were the other two overriding criteria that led to the emerging international network of ex situ base collections. The political framework for the conservation of plant genetic resources finds its roots in the International Undertaking of the FAO and became ‘turbulent rapid’ with the conclusion of the Convention on Biological Diversity. This paper reviews the history of the global ex situ conservation system with a focus on the international network of base collections. It assesses the major ex situ conservation approaches and methods with their strengths and weaknesses with respect to the global conservation system and highlights the importance of combining in situ and ex situ conservation.

## 1. Introduction

Plant genetic resources are the foundation of our food production system, thanks to the genetic diversity they contain. It is this genetic diversity, both between and within crop species and their wild relatives, that allow crops to evolve and adapt to changing conditions, either natural or human-created conditions. Since the first steps of early farmers to start the process of domesticating species from wild plants in the Near East more than 10,000 years ago, plant genetic resources and their diversity allowed humankind to develop crops according to its needs and to spread them around the world; thus, securing our plant food basis. 

Since these ancient times, the number of domesticated crops has steadily increased, and the cultivated forms or varieties of most of these crops have also increased and collectively become more diverse when moving from one region to another. Human and natural selection have been the driving force behind this diversification, but this process was only possible because of the genetic diversity available within and between these crop varieties and the related wild species that collectively form the diversity gene pool [1]. Genetic mutations in the crop genome are a permanent source of genetic diversity that allowed and continue to allow human and natural selection to be successful. Human exploitation of genetic diversity drastically increased when plant breeding became established, some 150 years ago [2]. This process of purposely generating new diversity through crossing different individuals followed by subsequent selection, resulted in high(er) yielding elite varieties. The success of this human managed evolution meant a steady replacement of older and usually well-adapted cultivars and even of entire crops. The loss of genetic diversity is called genetic erosion and was the trigger for targeted conservation efforts worldwide [3].

With the steady and increasing loss of genetic diversity since the middle of the last century for many of the crops cultivated worldwide and particularly for the main food crops, the need for systematic collecting and conservation of this diversity was recognized, and global conservation activities were initiated. Gradually, the Food and Agricultural Organization of the United Nations (FAO) in Rome assumed a coordinating role, supported by the International Board for Plant Genetic Resources (IBPGR), founded in 1974, one of the CGIAR centres, whose secretariat was initially based at FAO, thus serving as a technical and advisory institute for FAO and its political bodies such as the Commission on Plant Genetic Resources and later the International Treaty on Plant Genetic Resources for Food and Agriculture (ITPGRFA). Gradually, through IBPGR’s research, coordination and scientific advice and training were provided to countries worldwide, and a global network of plant genetic resources conservation centres, called genebanks, was established [4]. Political debates at FAO, and IBPGR’s research efforts aimed at collecting and improving the conservation of plant genetic resources for food and agriculture (PGRFA), somehow led to a more or less spontaneous creation of a global long-term conservation system of PGRFA [5]. This system underwent an evolutionary process itself, taking advantage of new scientific and technical developments and adjusting to evolving political conditions. It is this system that we will critically assess and review, with its strengths and weaknesses, with the aim to provide a perspective on how the system can be strengthened and can be made more rational in order to enable effective and efficient long-term conservation.

### 1.1. Scope

Whereas natural and human made ecosystems harbour the biodiversity of plants, animals, and microbes embedded in a physical environment, the focus in this paper will be on the plant genetic resources that are used for food and agriculture, i.e., PGRFA, or plant agrobiodiversity. These PGRFA comprise landraces and primitive and obsolete cultivars, crop wild relatives and modern varieties. Sometimes, plant breeding and other research materials are also regarded as genetic resources that should be included in genebanks.

Regarding the conservation activities, the main focus of this paper will be on long-term ex situ conservation, i.e., genebanks that manage seed, field, in vitro, and cryopreserved collections as well as DNA samples. Thus, not only are seeds important organs for conservation, but entire plants, pollen, tissues, cell suspensions, and more recently, DNA are also used. As not all plant agrobiodiversity can be collected and stored in genebanks, e.g., many wild food plants, many crop wild relatives, etc., we also look at in situ or nature conservation as well as at the on-farm maintenance of landraces and other genetic resources that require keeping the population structures of the material to be protected intact and/or to ensure a continuous evolution or the maintenance through steady cultivation or management. This dynamic evolutionary conservation stands in contrast to the frozen and static conditions that genebanks practice. Whereas ex situ conservation tends to focus on genotypes, in situ and on-farm conservation aim at natural and/or human-made populations and mixtures. It might be obvious that a balanced integration of these different conservation approaches will be needed to optimize the conservation system, as these approaches are complementary.

As conservation is frequently undertaken with the aim of keeping genetic diversity available and easily accessible for use, i.e., by farmers, breeders, or researchers, availability aspects are also important to be considered when deciding on the conservation ‘approach’. Therefore, due attention will be given to how to increase the use of materials conserved under long-term conservation conditions.

Detailed knowledge of the conserved genetic resources is a key requirement for rational, effective, and efficient conservation as well as to facilitate the use of the resources. Thus, research on plant genetic resources in situ or in genebanks is an essential activity to support these requirements. This aspect will be given due attention.

Besides the importance of creating new knowledge of the materials under conservation and to facilitate their use, the application of new technologies in conservation and use is critically important to achieve rational, efficient, and effective long-term conservation and to facilitate the use of plant agrobiodiversity.

Plant agrobiodiversity is distributed across the world; therefore, the sovereignty of national states is an important legal aspect that was recognized in the Convention on Biological Diversity (CBD), and the accessibility to these resources is thus determined by individual states. Moreover, genetic resources might be protected by intellectual property rights, hence the legal and policy framework for the conservation and the use of PGRFA is an important element to ensure rational, efficient, and effective conservation and use.

Other aspects that might directly or indirectly impact conservation decisions include training and capacity building, awareness creation, participatory approaches, economic considerations, and possibly others. These aspects are not the focus of this paper or of this Special Issue but can be of critical importance to achieve a rational and sustainable long-term conservation system.

### 1.2. Focus of This Review

In this paper, we will address the above-mentioned aspects or considerations of a long-term conservation system that might directly or indirectly impact the efficiency as well as effectiveness of the conservation and the facilitation of use of plant agrobiodiversity in all of their dimensions. A history of the (long-term) plant agrobiodiversity conservation developments will be presented to understand the ‘evolution’ of the system and its elements, also in the context of technical, scientific, economic, and social developments.

Brief descriptions of main conservation methods and the underlying concepts as well as of the main ex situ germplasm collection types are intended to provide a solid foundation for their critical review, as these are components of the FAO Global System for the conservation and sustainable use of plant genetic resources for food and agriculture (hereinafter called the global system) that has emerged over the past few decades. A useful definition of the ‘global system’ [6] refers to the worldwide community of genebanks and institutes that are working together and individually to conserve and use plant genetic resources for food and agriculture and the policy instruments and global action plans that bind them together and support their work. CGIAR genebanks, given the size and diversity of their collections, their global mandate, and the extensiveness of their partnerships form the central pillar of this system. Closely related to the previous points and possibly a conclusive statement is the need for complementary conservation approaches.

## 2. History of the Development of the Long-Term Conservation Practices and the Evolving Global Conservation System

Crop and related genetic diversity underpin the productivity, sustainability, resilience, and adaptive capacity of agricultural systems and, thus, their evolutionary potential [7]. This diversity, contained in the so-called plant genetic resources has played a key role in the developments of agriculture since the first steps towards the domestication of our crop plants, the subsequent diffusion of the domesticates as well as the associated weeds and wild relatives from the centres of domestication into the world and the ongoing improvement and adaptation of the crops to ever changing environments, cultural practices, and human-made and natural threats. The first farmers started to migrate out of the Fertile Crescent to new geographic areas about 10,000 years ago, carrying genetic resources with them [8]. Whereas this process will have caused bottlenecks and thus might have impacted the evolution of these crops, the introduction of new and possibly more genetic diversity, natural mutations as well as natural and human selection have resulted in an enormous diversity of crops and varieties. This traditional crop development process underwent significant changes through rediscovery, around the turn of the 20th century, of the laws of inheritance proposed by Gregor Mendel in 1865 and 1866, which formed the basis for the science of genetics and thus, the birth of scientific plant breeding [9].

One of the first persons to realize the importance and use the power of genetic diversity in crop improvement was Nicolai Vavilov, a Russian geneticist and a director of the Lenin All-Union Academy of Agricultural Sciences at Leningrad (now the Vavilov Federal Research Centre of Plant Genetic Resources—VIR) who was requested by Lenin, the head of the government of Soviet Russia and later the Soviet Union, to breed plants that could be cultivated in Siberia and thus would contribute to increased food production after the First World War [10]. Collecting about 50,000 samples of crop plants systematically and throughout the world and evaluating them to assess their traits, he realized that the collected genetic diversity was largely confined to restricted areas, the so-called centres of diversity/origin of our crops [11].

Plant introduction centres that later grew out into genebanks were established in several countries to meet the increasing demand by plant breeders for more diversity. These included the All-Union Institute for Plant Industry in St Petersburg (in 1920), the Commonwealth Potato Collection in Cambridge, UK (before the Second World War), collections for the research programmes of the Rockefeller Foundation in the USA (1943), and The National Seed Storage Laboratory (NSSL) in Fort Collins, CO, USA (1958) [12]. The latter became the long-term storage facility for valuable germplasm propagated by seeds from the four regional plant introduction stations and an inter-regional station for potato [12]. During the 1950s and 1960s, several national plant introduction centres/genebanks were established on all continents, plant quarantine regulations were initiated (such as those in West Africa), and plant exploration and collecting started (such as the initiatives in Latin American countries). During the 1940s and 1950s, advanced and well-organized global germplasm collecting missions were coordinated by the Rockefeller Foundation in the USA [12].

With the increasing successes of plant breeding and the spread of modern and frequently high-yielding varieties, especially of the major food crops, a process of variety and later, even a process of crop replacement started and resulted in significant losses of genetic diversity, a development that was called ‘genetic erosion’ [13]. As early as 1936, Harlan and Martini raised the issue of genetic erosion in a USDA report devoted to barley breeding [14], and Vavilov had noted the increased loss of landraces. Particularly, during the so-called ‘Green Revolution’, which started in the late 1950s until the early 1970s, the success of high-yielding (dwarf) varieties of wheat and rice, together with new agricultural technologies, led to drastic losses of the traditional landraces of these crops, and this triggered concern in organizations such as the European Society for Research and Plant Breeding (EUCARPIA) and FAO [12]. In 1966, the EUCARPIA delegates advised European plant breeding institutes to foster continental collaboration through the establishment of four sub-regional genebanks in what was then West Germany (FAL in Braunschweig, for NW Europe); in East Germany (Gatersleben), Poland (Radzikow), Russia (St Petersburg) and/or others for Central and Eastern Europe; in Italy (Bari, for Southern Europe); and Sweden (Lund, for the Nordic countries) [12]. Gradually, regional and global networking increased, and the contours of a global conservation system became visible.

### 2.1. The Role of FAO

During the 1950s and early 1960s, FAO became the major actor in the conservation of plant genetic resources. Besides the World Catalogues of Genetic Stocks for wheat, rice, maize, and barley, they started to publish the FAO Plant Introduction Newsletter and organized technical meetings/conferences (see below). Salient historical events with respect to the global conservation system are summarized in Table 1 and, where applicable, reference to the Table is made in the text. The first meeting was called the ‘Technical Meeting on Plant Exploration and Introduction’ and was held in 1961 (Table 1) [15]. A Panel of Experts on Plant Exploration and Introduction was established in 1965. The panel included visionary scientists such as Sir Otto Frankel (CSIRO, Australia), professor Jack Harlan (University of Illinois, Urbana, IL, USA), and Professor Jack Hawkes (University of Birmingham, Birmingham, UK); in addition, Ms. Erna Bennett, (FAO, Rome, Italy) served as one of the supporting secretarial staff members of the panel. Reports of the six panel meetings were published between 1968 and 1974 [16]. This panel also played an important role in the planning and steering of the first two International Technical Conferences that the FAO organized in collaboration with their partners [17].
The first International Technical Conference was held in 1967 in Rome and was organized by FAO and the International Biological Programme (IBP) under the title ‘Technical Conference on the Exploration, Utilization and Conservation of Plant Genetic Resources’ (Table 1) [18]. Some of the major recommendations of the 1967 conference included the need to survey genetic resources in nature and in genebanks and the need for a stronger emphasis on conservation, efficient documentation, and the improved international coordination of PGR activities. It also generated important guidelines for the establishment of a global network for ex situ long-term conservation. It should also be noted that in situ conservation, especially of landraces, was a big issue, but it was given little to no importance compared to ex situ conservation [12,13].In 1971, the second international conference on crop genetic resources was held in Rome, and its proceedings were published in the book *Crop Genetic Resources for Today and Tomorrow*, which included a plan of action (Table 1) [19]. At this conference, the panel of experts made some major contributions with respect to global conservation plans, including the formulation of basic criteria for the conservation and the use of genetic material. These were: (i) that plant material was to be made available immediately and without restriction to all breeders requesting it and (ii) that genetic variability had to be maintained for future generations in long-term storage under conditions for maximum physical and genetic security. A third important result of the panel was a categorization of ex situ collections: base collections (for long-term conservation), active collections (for research and distribution), and working collections (usually maintained at plant breeding institutions) (for details, see [20]. They also identified regions and crops for priority collecting [3]. These collecting priorities were reformulated during the panel’s last meeting in 1975, with a clear shift from crops to regions [13].The third international conference on crop genetic resources was held in Rome in 1981, jointly organized by FAO, UNEP, and IBPGR (Table 1) [21]. The conference addressed most of the routine genebank operational topics, including sampling, seed storage and viability monitoring, recalcitrant seeds, in vitro conservation and the genetic stability of cultures, principles of germplasm regeneration, in situ conservation, the use of back-garden and genetic reserves for regeneration, the principles and practice of germplasm distribution and exchange, the safe and rapid transfer of plant genetic resources, including a proposal to distribute only germplasm materials completely free from plant pests and pathogens, principles of characterization and evaluation, data capturing and germplasm documentation, and under-exploited and minor crops [21].The fourth technical conference was in the context of the FAO global system for the conservation and use of plant genetic resources and was held in Leipzig, Germany in 1996 (Table 1) [22]. The major outcome of this conference was the Global Plan of Action (see below) and, in addition, ample information on the global conservation system [22].

The rising concern regarding the genetic erosion of landraces and wild relatives due to modern agriculture, and the more general, increasing need of the agro-industry for a steady flow of new germplasm convinced the members of the FAO conference to give more consideration to a generalist approach to conservation [12]. During the second conference, the availability of new cold-storage techniques was noted, thus allowing long-term ex situ storage to be undertaken, whereas advocated in situ conservation, based on genecological premises, did not materialize until much later. The focus remained on ex situ conservation, despite the arguments for in situ approaches [3].

It should be noted that during the 1960s, the discussions on PGR in general as well as within FAO were dominated by plant breeders, and this resulted in a close conceptual link between conservation and use. Moreover, germplasm was predominantly stored in industrial countries and was closely tied to plant breeding institutes. During 1967, the FAO unit of Crop Ecology and Genetic Resources was established and thus provided FAO with more in-house specialized expertise.

### 2.2. The Establishment of the International Board for Plant Genetic Resources (IBPGR)

During a meeting of the Technical Advisory Committee (TAC) of the CGIAR in Beltsville, USA, a group of invited external experts, including several members of the FAO panel of experts, presented an ambitious plan to establish a world network of genetic resources centres [40]. This plan consisted of four elements. The first one was to establish a coordinating centre (to become IBPGR); the second one was to stimulate the establishment of genebanks in already existing international centres in developing countries (i.e., IRRI, established in 1960; CIMMYT (1966); CIAT (1967); and IITA (1968). The third element was to establish genebanks in new international centres (WARDA, 1971; CIP, 1971; and ICRISAT, 1972). Soon thereafter, the ILCA was established in 1974, and ICARDA was established in 1976. The fourth element was the establishment of new ‘regional’ centres in the Vavilovian centres for crop diversity. The establishment of the International Board for Plant Genetic Resources (IBPGR) took place in 1974, as a secretariat for its board of trustees, administered by FAO and technically as one of the international institutes of the CGIAR. It was expected to coordinate global exploration and collecting efforts and to orchestrate a global network of genebanks (see also the details of this international undertaking below). Its main task was formulated as ‘*to promote and assist in the worldwide effort to collect and conserve the plant germplasm needed for future research and production*’ [40].

The main achievements of IBPGR and its successor institute IPGRI, particularly those related to long-term conservation and the global conservation system, are updated from a list in [13] and include:Organization of collecting missions, partly using consultants in addition to its own staff and through contracts with national (selected) genebanks (for details, see IBPGR Annual Reports, e.g., [41]; for an overview: [42,43].Support for national and regional PGR programmes, predominantly in developing countries with the establishment of conservation facilities, documentation systems, and capacity building/training [41].Establishment of regional and global PGR networks with national programmes as principal stakeholders as well as regional and global crop networks, frequently with and through CGIAR centres and their leading roles in crop specific conservation and breeding, thus trying to ensure a close link between conservation and use. The European Cooperative Program for Plant Genetic Resources (ECPGR), formerly the ‘European Cooperative Programme for Crop Genetic Resources Networks’—ECP/GR), was founded in 1980 on the basis of the recommendations of the United Nations Development Programme (UNDP), the Food and Agriculture Organization of the United Nations (FAO), and the Genebank Committee of the European Association for Research on Plant Breeding (EUCARPIA); its secretariat was hosted by IBPGR [44].The establishment of an international network of base collections in 52 selected genebanks located in almost 40 countries across all continents for the long-term conservation of crops or crop groups, including 80 genera and approximately 250 species [45], and the so-called Registry of Base Collections containing a total of 144,000 accessions [43].Support for an international MSc course in the conservation and use of PGR at the University of Birmingham and the organization of training courses [41].Establishment of a digitalized information system for genebank documentation and germplasm management.Initiating, coordinating, and/or conducting plant genetic resource conservation and use research and publishing the results and procedures.More recently, the successor institutes of IBPGR (IPGRI and Bioversity International), especially after their administrative separation from FAO, played an active role in developing legal and policy proposals and acted as the CGIAR representative in international meetings and activities.

### 2.3. The International Undertaking (IU)

The International Undertaking (IU) was established by the FAO Commission on PGR in 1983 as a non-binding intergovernmental agreement to promote the conservation, exchange, and use of plant genetic resources [27]. Its objective was to ensure that plant genetic resources of economic and/or social interest, particularly for agriculture, would be explored, preserved, evaluated, and made available for plant breeding and scientific purposes. The Undertaking was based on the universally accepted principle that plant genetic resources are a heritage of mankind and, consequently, should be available without restriction. It defined ‘*plant genetic resources*’ *as the reproductive or vegetative propagating material of the following categories of plants: (i) cultivated varieties (cultivars) in current use and newly developed varieties; (ii) obsolete cultivars; (iii) primitive cultivars (landraces); (iv) wild and weedy species, near relatives of cultivated varieties; (v) special genetic stocks (including elite and current breeder lines and mutants)*. It defined ‘*base collection of plant genetic resources*’ *as a collection of seed stock or vegetative propagating material (ranging from tissue cultures to whole plants) held for long-term security in order to preserve the genetic variation for scientific purposes and as a basis for plant breeding*; ‘*active collection*’ was defined as ‘*a collection which complements a base collection, and is a collection from which seed samples are drawn for distribution, exchange and other purposes such as multiplication and evaluation*’, and ‘*centre*’ *was defined as an institution holding a base or an active collection of plant genetic resources* [46].

Furthermore, the IU foresaw the development of a global system as to ensure that (Article 7.1):A well-coordinated international network of national, regional, and international genebanks, including the international network of base collections, would develop. The unrestricted availability of materials included in the active and base collections of such a network was assumed.Through the progressive growth of the network, a comprehensive coverage of species and regions was aspired, and an adequate safety duplication of the germplasm was involved.The exploration, collection, conservation, maintenance, rejuvenation, evaluation, and exchange of plant genetic resources should be conducted by the genebanks in accordance with scientific standards.Adequate funding should be provided.A global information system should be developed.Genebanks should give an early warning to the FAO in the case of hazards that threaten the efficient maintenance of the collection.IBPGR is expected to liaise with FAO while conducting its programme of work aiming at building institutional and human capacity within developing countries for the development and distribution of improved crop varieties.

Article 7 of the IU on International Arrangements addresses aspects of the global system and access to germplasm in the base collections. Countries are invited to notify the FAO in case their base collections are to be recognized as part of the international network of base collections. The participating genebanks are expected to make the materials in these base collections available to the participants in the IU for the purposes of scientific research, plant breeding, or conservation, free of charge and based on mutual exchange or mutually agreed terms [46].

The IU was replaced by the International Treaty on Plant Genetic Resources in 2002 (see further below).

Another component of the global system is the International Code of Conduct for Plant Germplasm Collecting and Transfer [47]. It was adopted by the FAO Conference at its 27th session in 1993. The voluntary code aims to promote the rational collecting and sustainable use of genetic resources to prevent genetic erosion and to protect the interests of both the germplasm collectors and donors. It is based on the principle of national sovereignty over PGR and is in harmony with the CBD [47]. 

### 2.4. The Convention on Biological Diversity (CBD)

The negotiation of the Convention on Biological Diversity (CBD) in the eighties and early nineties, under the auspices of the United Nations Environment Programme [48], did result in drastic changes with respect to the conservation and use of PGRFA. Besides creating a general, globally, and legally binding framework for the conservation and sustainable use of biodiversity, the CBD, which entered into force in 1993, required that access to valuable biological resources must be conducted on ‘mutually agreed terms’ and is subject to ‘prior informed consent’ of the country of origin. The national sovereignty of states over biodiversity within their borders was recognized as a key principle in the CBD, and consequently, this became the ‘driving force’ in the thinking and approaches to the negotiations and future developments. Besides the fact that states were expected to ‘look after their own biological resources and conserve them, whenever possible in their own country’, this also caused a strong incentive for countries to favour bilateral rather than multilateral arrangements for the exchange of genetic resources. 

From an agricultural perspective, it should be noted that the negotiations of the CBD were strongly influenced by environmentalists and nature conservationists and, consequently, a bias towards wild (i.e., non-domesticated and non-agricultural) plant and animal species could be observed [49]. In fact, agriculturalists were hardly present in the negotiations, and it was only through a separate resolution (Resolution 3 of the Nairobi Final Act) [30] that the FAO was asked to address two important but unresolved agricultural genetic resources issues, i.e., the question of Farmers’ Rights and the need to address the legal status of existing genetic resource collections established prior to 1993 [50].

The negotiation process of the CBD caused a dramatic shift concerning the overall conservation approach, i.e., from a rather technologically driven ex situ conservation approach (‘putting the germplasm safely away for the future’), towards a much more people-centred conservation, with a strong emphasis on in situ and on-farm conservation and sustainable use efforts. Alongside this, due attention was being paid to participatory research (and conservation) activities to recognize the important role of local communities in the management of and their dependency on biodiversity. This also led to the recognition of traditional and indigenous knowledge to be an important component of biodiversity that needs to be collected and/or conserved. The importance of technology for the conservation and use of genetic resources should be recognized as well as the provision of access to such ‘enabling’ technologies. These aspects facilitated (and required) a much closer link between conservation and development and led to a greater participation of local communities and subsistence farmers in conservation and use related activities. It is against this background that the access and benefit-sharing guidelines were developed and agreed upon in 2002 within the framework of the CBD by an Ad Hoc Open Ended Working Group on Access and Benefit-Sharing [51] that eventually, in 2010, resulted in the adoption of the Nagoya Protocol on Access and Benefit Sharing (ABS), which entered into force in 2014 [52]. It is a supplementary agreement to the CBD convention of 1992 and aims at the implementation of one of the three objectives of the CBD: the fair and equitable sharing of benefits arising out of the use of genetic resources, thereby contributing to the conservation and sustainable use of biodiversity [52]. Its rather strong focus on wild species and the bureaucracy involved to apply the protocol have resulted in concerns that the added bureaucracy and legislation could be damaging to the monitoring and collecting of biodiversity, to conservation, and to research, because the protocol severely limits access to genetic resources.

The CBD recognizes the application of intellectual property rights (IPRs) on biological materials as a means of protecting inventions and stimulating innovation. This led to a further expansion of the scope and/or application of IPRs, especially patents and plant breeder rights (PBRs), in agricultural research and plant breeding. Due to concerns that the development and use of genetically modified varieties could cause a threat to the environment and its biological resources, a legal framework on biosafety aspects was demanded, and thus, the so-called Cartagena Protocol on Biosafety was developed and came into force in 2003 as a legal framework for biosafety legislation and is yet another supplementary agreement of the CBD [53].

At present, the negotiation process on the development of the post-2020 global biodiversity framework is ongoing for its adoption during the forthcoming meeting later in 2021 in Kunming, China [54].

### 2.5. Global Plan of Action (GPA)

The first Global Plan of Action (GPAI) for conserving and using crop diversity was adopted in 1996 by 150 countries [22]. The GPAI called for a rational global conservation system based on the principles of effectiveness, efficiency, and transparency. The Second Global Plan of Action (GPAII) reiterated that call and provided a strategic framework for the conservation and the sustainable use of plant genetic diversity. It was adopted by the FAO Council in November 2011 and reaffirmed the commitment of governments to the promotion of plant genetic resources as essential components of food security through sustainable agriculture in the face of climate change (Table 1) [39]. It is a rolling action plan and is based on the findings of the Second Report on the State of the World’s PGRFA [38] and inputs from a series of regional consultations and from experts. The GPAs are a supporting component of the International Treaty on Plant Genetic Resources for Food and Agriculture [55].

The GPAII consists of four main groups of priority activities, i.e., in situ conservation and management, ex situ conservation, sustainable use, and building sustainable institutional and human capacities [39]. The in situ conservation group of four priority activities comprises: 1. surveying and inventorying PGRFA; 2. supporting on-farm management and improvement of PGRFA; 3. assisting farmers in disaster situations to restore crop systems; and 4. promoting in situ conservation and management of crop wild relatives and wild food plants. The ex situ group of priority activities includes: 5. the targeted collecting of PGRFA; 6. sustaining and expanding ex situ conservation; and 7. regenerating and multiplying ex situ accessions. The sustainable use priority activities consist of: 8. the characterization and evaluation and development of subsets of collections to facilitate use; 9. plant breeding, genetic enhancement, and base broadening; 10. promoting the diversification of crop production and broadening crop diversity; 11. the development and commercialization of varieties, primarily of farmer varieties/landraces and underutilized species; and 12. supporting seed production and distribution. The set of capacity building activities comprises: 13. building and strengthening national programmes; 14. promoting and strengthening networks for PGRFA; 15. constructing and strengthening comprehensive information systems; 16. developing and strengthening systems for monitoring and safeguarding genetic diversity and minimizing genetic erosion of PGRFA; 17. building and strengthening human resource capacity; and 18. promoting and strengthening public awareness of the importance of PGRFA [39].

The GPAII does not contain specific activities related to long-term conservation and the global system, but several comments and supporting actions are referred to throughout the text, e.g., that the network of international ex situ collections of major crops played an important role in the negotiations of the International Treaty. These collections continue to form the backbone of the global system. The Svalbard Global Seed Vault now provides an additional level of security to existing ex situ collections [37]. Furthermore, the development of a global portal of accession-level data and the imminent release of an advanced genebank information management system (recently released and called GLIS) are additional important steps towards the strengthening and more effective operation of a global system for ex situ conservation [56]. Enhancing capacity at all levels is a key strategy to implement the priority activities of the GPA, including those related to long-term conservation, sustainable use (i.e., plant breeding, genetic enhancement, and base-broadening efforts) and the global system. Whereas countries have national sovereignty over and responsibility for the PGRFA they conserve, there is nevertheless a need for the greater rationalization of the global system for ex situ collections. The fostering of partnerships and synergies among countries is a requirement to develop a more rational and cost-effective global system. Furthermore, the GPAII plays an important role in the international policy framework for world food security and as a supporting component of the International Treaty. It contributes to achieving the Millennium Development Goals and aids in the implementation of the Strategic Plan for Biodiversity [57].

### 2.6. International Treaty on Plant Genetic Resources for Food and Agriculture (ITPGRFA)

The International Treaty for Plant Genetic Resources for Food and Agriculture (ITPGRFA or Treaty) aims to recognize the enormous contribution of farmers to the diversity of crops that feed the world; it aims to establish a global system to provide farmers, plant breeders, and scientists with access to plant genetic materials; and it aims to ensure that recipients share the benefits they derive from the use of these genetic materials with the countries where they originated [55]. The preparations and negotiations of the revision of the IU were initiated in 1994 and were concluded in 2001 by the adoption of the International Treaty. It encompasses all PGRFA and came into force in 2004 [55].

Through the Treaty, countries agree to promote the development of national integrated approaches to the exploration, collecting, characterization, evaluation, conservation, and documentation of their PGRFA, including the development of national surveys and inventories [55]. They also agree to develop and maintain appropriate policies and legal measures to promote the sustainable use of these resources, including on-farm management, strengthening research, promoting plant-breeding efforts, broadening the genetic bases of crops, and expanding the use of locally adapted crops and varieties and underutilized species. These activities would be supported, as appropriate, by international cooperation provided in the Treaty.

The most important part of the ITPGRFA is the establishment of the so-called Multilateral System (MLS) of Access and Benefit-Sharing [58]. The MLS applies to 64 genera, including the major food crops and forages, which were agreed upon on the basis of two criteria: their importance for food security and the level of interdependence among countries. At the global level, these crops provide approximately 80% of the food that is produced by plants. Through the MLS, sovereign nations have agreed to share resources and benefits. The genetic resources included in the MLS will be made available for research, breeding, and training, and their recipients should not claim any intellectual property or other rights that limit access to these resources or their genetic parts or components in the form received from the MLS [59]. The peculiarities of PGRFA compared to biodiversity in general, e.g., the difficulty of applying the country-of-origin concept, the strong interdependency of nations on genetic diversity for crop improvement, and the critical role of these resources in traditional agriculture and in food security, formed the basis for the establishment of a multilateral rather than a bilateral system for their exchange [60]. This thinking eventually led to the establishment of the MLS, which keeps the genetic resources of the Annex 1 listed species that are formally in the public domain and under governmental control and facilitates easy access to and the use of these resources [49]. It should be noted that the diversity of the crop species or the groups of species listed in Annex I is rather limited and, for instance, the majority of the vegetable genetic resources conserved by the World Vegetable Center in Taiwan, which consist of a large proportion of indigenous vegetables that are critically important for the diversification of cropping systems, nutritional security, and livelihoods [61], are not included in Annex 1. Discussions on the extension of the Annex 1 list have been ongoing for several years, but no final decision has been reached.

The benefits arising from the use of materials from the MLS shall be shared fairly and equitably through the exchange of information, access to and transfer of technology, and capacity-building, considering the priority activity areas indicated in the above mentioned GPAII and under the guidance of the Governing Body of the Treaty. It further establishes the payment, which is in certain cases mandatory, of an equitable part of the monetary benefits that are derived from the use of PGRFA into the funding strategy of the Treaty [58]. The funding strategy aims at mobilizing funds for activities, plans, and programmes to support the implementation of the Treaty and, in particular, its implementation in developing countries while keeping in line with the priorities that have been identified in the GPA. The funding strategy includes the monetary benefits that are paid in accordance with the MLS as well as the Global Crop Diversity Trust, which is described below. The Treaty recognizes the enormous contributions that local and indigenous communities and farmers of all regions of the world have made and will continue to make for the conservation and development of PGRFA. The Treaty makes governments responsible for the realization of Farmers’ Rights, including the protection of relevant traditional knowledge; provisions for farmers to participate equitably in sharing benefits; and farmer participation in national policy decision-making [55,59]. Through Article 15, the Treaty establishes its relationship with the CGIAR and other international centres: ‘*Ex Situ Collections of Plant Genetic Resources for Food and Agriculture held by the International Agricultural Research Centres of the Consultative Group on International Agricultural Research and other International Institutions*’ and arranges that the materials listed in Annex 1 of the Treaty and that are held by the centres as well as other species than those listed in Annex 1 of this Treaty and collected before its entry into force that are held by IARCs shall be made available in accordance with the provisions of the standard material transfer agreement (SMTA) [55].

### 2.7. International Network of Ex Situ Collections

The international network of ex situ base collections in genebanks that are managed by national, regional, or international centres was a component of the section on the international arrangements of the International Undertaking. It was foreseen that through a steady increase of the number of genebanks participating in the network, adequate coverage in terms of species and geographical distribution would eventually be achieved. It was further foreseen in the IU to conclude agreements (four ‘model agreements’ were available to choose from) with countries to place their base collections within this network and/or to provide storage space for the long-term storage of base collections from elsewhere. A few countries and institutions made concrete offers to place (part of) their collections in the network. The latter would operate under the auspices and/or the jurisdiction of the FAO and a number of contracts were concluded (see below).

In 1994, the CGIAR centres expressed the wish that their designated germplasm be recognized as part of the international network of ex situ collections and signed individual agreements with FAO [62]; Chapter 3.1. in [38]. The salient features of these agreements based on one of the above-mentioned model agreements include that:The centre shall hold the designated germplasm in trust for the benefit of the international community.The centre shall not claim legal ownership over the designated germplasm, nor shall it seek any intellectual property rights over that germplasm or its related information.The designated germplasm shall remain in the charge of the centre.The FAO shall have a right of access to the premises at any time and has the right to inspect all activities performed therein.The centre shall undertake the management and the administration of the designated germplasm in accordance with internationally accepted standards with respect to the storage, the exchange and distribution of seeds, the international genebank standards endorsed by the Commission and that all designated germplasm is duplicated.The centre recognizes the intergovernmental authority of the Commission in setting policies for the International Network and shall undergo consultation with the FAO and its Commission on proposed policy changes related to the conservation of the germplasm.The centre shall undertake the creation of samples of the designated germplasm and will make related information available directly to users or through the FAO for the purposes of scientific research, plant breeding, or genetic resource conservation without restriction.The centre shall ensure that such other people or institutions and any further entity receiving samples of the designated germplasm from such a person or institution are bound by the conditions to not claim ownership over the materials or to seek any intellectual property rights over that material and, in the case of samples duplicated for safety purposes, to manage these in accordance with internationally accepted standards.

A related network, as mentioned above, is the so-called Register of Base Collections that was established by the IBPGR in the 1970s and includes genebanks that were prepared to accept a long-term commitment to conserve germplasm materials and to make these available to users. This register formed the backbone of the international network of base collections. For details, see the paper by Engels and Thormann [42].

It should be noted that further agreements have been concluded with several other international research centres (e.g., the World Vegetable Center, CATIE and CRU, and some regional organizations (e.g., South Pacific Community)). Agreements with individual countries have not been vigorously pursued. In October 2006, 11 CGIAR centres signed agreements with the Governing Body of the International Treaty to bring their in trust collections under the framework of the Treaty and to recognize the authority of the Governing Body providing policy guidance related to those collections [63,64].

With the establishment of the International Treaty and its Multilateral System, the network of ex situ collections, and the conclusion of the agreements with the centres of the CGIAR, these collections were brought under the International Treaty (Chapter 3.2 in [38]). The commitments of countries to conserve germplasm for the long-term and to make the materials available (under an SMTA) have been made by countries and genebanks through the inclusion of germplasm in the MLS.

### 2.8. The Institutional and Capacity Building Framework

The establishment of the IBPGR has already been mentioned above, as it was intricately linked to political debate and developments during the 1970s (see Section 2.2). Similarly, the other centres of the CGIAR that operate genebanks with the genetic resource collections of their respective mandate crops are important elements of the emerging global ex situ conservation system. Since its establishment, the IBPGR has played an active role in strengthening this global system by supporting national PGRFA programmes and facilitating the establishment of new regional genebanks as part of the global network. In 1976, the formation of regional programmes in Southeast Asia and Europe and the establishment of (regional) genebanks in Costa Rica and Ethiopia (with funding from Germany) as well as the support provided to students from developing countries to attend the MSc programme on plant genetic resources at the University of Birmingham was reported [41]. Furthermore, the annual report listed international and regional institutions that accepted the invitation to become the holders of ‘world’ base collections of crops of global importance. During the following years, a steady increase of arrangements for regional programmes was reported as well as the development of a computer-based information and retrieval system, support provided to establish or strengthen national programmes and training activities as well as the acceptance of recommendations on the physical and engineering design of long-term seed stores [65,66,67].

### 2.9. Global Crop Diversity Trust

The Global Crop Diversity Trust (Crop Trust) was established in October 2004 by the IPGRI, now Bioversity International, on behalf of the CGIAR and FAO to help support the global system in a sustainable way through a Crop Diversity Endowment Fund [35]. Its mission is to ensure the conservation and availability of crop diversity for food security worldwide. Among others, the Trust provides oversight of the CGIAR Genebank Platform. The 11 CGIAR genebanks safeguard a unique global resource of crop and tree diversity and respond to thousands of requests for germplasm from users in more than 100 countries worldwide every year [68]. The goal of the CGIAR Genebank Platform is to conserve these collections and to make this diversity available to breeders and researchers in a manner that meets international scientific standards and that is cost-efficient, secure, reliable, sustainable over the long-term and that is supportive of the Plant Treaty. The Crop Trust has oversight over and financial responsibility for these CGIAR genebanks [69].

### 2.10. Some Critical Side-Effects on the Global Conservation System

The above-described developments had some significant (perceived?) side-effects on the emerging global long-term conservation system. They included a boost to the establishment of (national) genebanks, among others, triggered by the CBD’s recognition of national sovereignty. The acceptance of intellectual property rights over genetic resources resulted in a steady increase of access regulations to genetic resources. Furthermore, issues of ownership over genetic resources emerged, leading to the refusal of some countries to provide access to ‘their’ plant genetic resources. Against this backdrop, a rather legalistic thinking of access and benefit sharing developed and influenced the arrangements in this field of the International Treaty.

Evolving molecular and later genomic techniques allowed and facilitated the assessment of genetic diversity aspects, including the identification of duplicate accessions; a quantification of genetic diversity; the identification of alleles and genes and their functions as well as their transfer between individuals and species. A better understanding of genetic diversity also allowed for more targeted collecting, better characterization/evaluation, and greatly facilitated plant breeding. The creation of so-called GMO (genetically modified organism) varieties with the help of these new molecular and biotechnology tools became a ‘hot issue’, among others, due to their threat to the genetic diversity of crop germplasm collections and biodiversity hotspots [70], and this caused restrictions or even prohibition of related research or the cultivation of modified materials. The multilateral thinking became an ‘alternative’ to restricting ownership; more IPRs crept in and resulted in heavy debates and in more restrictive attitudes regarding sharing natural genetic resources. All of these developments and possible repercussions call for a critical review of the current global system as it has evolved in the context of the above-described developments and the mentioned side-effects to provide elements for the creation of a more efficient and rational system of global base collections of important food crops.

## 3. Description of Ex Situ Germplasm Conservation Methods and Their Strengths and Weaknesses

The vast majority (approx. 92%) of angiosperms comprising roughly 330,000 species of flowering plants has desiccation-tolerant and so-called orthodox seeds [71,72] that survive drying to a low moisture content, 5% or less, and subsequent rehydration without a significant loss of viability [73,74]. Orthodox seeds acquire desiccation tolerance during their late phase of development when they undergo pre-maturation drying and are later shed metabolically inactive [72]. Desiccation tolerance is lost during germination [75]. Moreover, most desiccation-tolerant species tolerate low temperature (sub-zero) storage and seed longevity increases, within certain limits, with a decrease in seed moisture content (SMC) and storage temperature [76]. Harrington [77] postulated two rules of thumb regarding seed longevity in storage that apply independently. Over the range of 14 to 4% SMC (fresh weight basis), a 1% reduction in SMC doubles the life span of the seed. Similarly, within the range of 50 to zero degrees Celsius, for each 5 °C drop in storage temperature, the life span of seed in storage would double. Therefore, the cold storage of dried seeds is a practical, efficient, and cost-effective method for the long-term storage of germplasm in genebanks. The FAO Genebank Standards recommend storage at −18 ± 3 °C and a relative humidity of 15 ± 3 percent for most original seed samples and safety duplicate samples intended for long-term storage [78]. In case seed samples are stored in hermetically sealed pouches or containers, the control of the storage room RH is not required.

In contrast to orthodox seeds, so-called ‘recalcitrant’ seeds are desiccation-sensitive and rapidly loose viability upon drying and do not tolerate low temperature storage [73]. Recalcitrant seeds undergo extremely limited drying during maturation and consequently, have high SMC and are metabolically active during shedding [79]. Desiccation sensitivity also seems to be linked to the non-dormant state of seeds upon shedding [71]. The SMC below which viability is lost varies between species but is generally above 20% [80]. Specifically, tree species of tropical provenance, such as avocado (*Persea americana*), cacao (*Theobroma cacao*), jackfruit (*Artocarpus heterophyllus*), breadfruit (*Artocarpus altilis*), lychee (*Litchi chinensis*), mango (*Mangifera indica*), mangosteens (*Garcinia mangostana*), etc., produce recalcitrant seeds.

There is a third category of seed storage behaviour comprising so-called ‘intermediate’ seeds without sharp boundaries between orthodox and recalcitrant seeds [81]. Species with intermediate seed storage behaviour can be dried to certain SMC levels but cannot be dried to a level as low as truly orthodox seeds [82] and often do not survive sub-zero storage temperatures. Moreover, seeds with intermediate storage behaviour tend to lose viability much quicker than orthodox seeds [82]. Coffee (*Coffea arabica*) seeds fall into this category of intermediate seeds [81]. Depending on the cultivar, coffee seeds tolerate drying to 5–10% SMC but viability at low or sub-zero temperatures is rapidly lost. Seeds of alpine species are also significantly shorter lived than their lowland counterparts, possibly due to abnormal seed development under the cool and wet conditions of the alpine climate [83].

As species producing seeds with intermediate or particularly recalcitrant storage behaviour have extremely limited longevity in a seed genebank, they are commonly stored in field genebanks and/or as in vitro collections for medium-term conservation and/or in liquid nitrogen for long-term conservation.

### 3.1. Short-, Medium- and Long-Term Ex Situ Storage of Orthodox Seeds

In general, orthodox seeds are relatively small and require little storage space for the conservation of a representative sample of the source population and further sub-samples for distribution, viability checking, and safety back-up. Crops commonly conserved in seed genebanks include cereals such as rice, wheat, barley, oats, sorghum, millet, maize, grain and forage legumes, most vegetables, and some fruit crops. True seeds of crops such as those from potato, which are commonly propagated vegetatively, can also be dried and stored at low temperature [84]. This is common practice with wild potato germplasm, and accessions are maintained as botanical seeds or true-potato seeds (TPS). A representative number of 20–50 individuals are typically collected from a wild population, and seeds are regenerated and combined to form a unique genebank accession of heterogeneous seed, which is expected to represent most alleles found in that population [85]. Seed samples of such wild potato germplasm accessions thus represent a heterogenous mix of genotypes, whereby each genotype represents a portion of the genetic make-up of the sampled population.

The core operations of a genebank conserving the seeds of orthodox species comprise cleaning, seed drying, viability and health testing, packing, storage, and distribution to users and for a safety back-up [86]. When seed stocks are running low or when seed viability drops below a minimum threshold, seed lots need to be regenerated for seed replenishment. All these genebank operational steps are documented and in many genebanks are supported by a genebank information system [87].

Most genebanks conserving PGRFA have the mandate to distribute germplasm to a range of different users and, for practical reasons, store the seeds of most collected or acquired accessions in a base and an active collection when justified. The most-original seed samples are kept in the base collection for long-term conservation, aiming at the highest level of genetic integrity of the stored sample with the original sample [78]. The active collection is oriented towards seed regeneration (triggered by low viability), characterization, evaluation, multiplication (triggered by low seed stock), and distribution and is generally kept under medium-term storage (MTS) conditions.

The base collection for any given species or a crop genepool may be distributed over several institutions, as is the case in Europe, with the implementation of a European Genebank Integrated System, abbreviated as AEGIS [88]. In contrast, the United States Department of Agriculture (USDA-ARS) has a network of genebanks holding the active collections for different crops in 19 different locations across the country, with one main base collection held at the National Laboratory for Genetic Resources Preservation (NLGRP) in Fort Collins, Colorado, serving all of the regional genebanks. The NLGRP maintains the US system backup of more than 445,000 accessions, representing 86% of the seed collections and 15% of the clonal collections [89]. Seeds are not distributed from the base collection to the users, but rather, they are distributed from the active collections.

The active collections comprising the bulk of orthodox seeds stored in most genebanks are to be kept under medium-term storage (MTS) conditions at temperatures ranging from 5 °C to 10 °C and at a relative humidity (RH) of 15 ± 3 percent for seeds that are stored in open containers [78]. Frequently, MTS conditions have a narrower range from +2 to +5 °C [86,89,90], and RH adjustment is not required if seeds are stored in hermetically sealed pouches or containers. Refrigerated seed storage under MTS conditions is adequate for up to 30 years [78]. It should be noted that seeds stored in hermetically closed containers are to be dried in a controlled environment with a temperature range between 5 and 20 °C and a RH between 15 and 25%, depending on the species.

The base collections are stored under long-term storage (LTS) conditions at sub-zero temperatures of typically −18 to −20 °C [86,89,90,91], and the seeds are dried as mentioned above for MTS, maintaining high seed quality over long, species-specific periods of up to 100 years or more.

Other genebanks whose major focus is not the use plant agrobiodiversity facilitation but rather whose focus is on the long-term conservation of globally threatened species (with relatively few sample requests), store all of their seeds exclusively under LTS conditions. This applies, for example, to the Millennium Seed Bank (MSB) of the Royal Botanic Gardens Kew, where dried seeds are transferred to air-tight glass containers or aluminum foil bags and are stored in the seed vault at −20 °C [91].

Assessing 42,000 seed accessions representing 276 species in the USDA National Plant Germplasm System provided evidence that some species produce orthodox seeds of short longevity in dry storage [92]. Some plant families had typically short-lived seeds (e.g., Apiaceae and Brassicaceae) or long-lived ones (e.g., Malvaceae and Chenopodiaceae). Moreover, environmental factors seem also to determine seed longevity, as seeds from species originating from certain localities in Europe had short shelf lives, while seeds of the same species originating from localities in South Asia and Australia had much longer shelf lives. For these reasons, some genebanks additionally cryopreserve samples of those orthodox seeds that are expected to be very short-lived, even under LTS conditions [93,94].

Under short-term storage (STS) conditions, the seed quality and the viability of orthodox seeds with long shelf lives can be maintained for a minimum of eight years under ambient conditions if 25 °C is not exceeded, and the relative humidity in the storage room is kept at 10–25% [78]. At the World Vegetable Center in Taiwan, working collections of breeders and other researchers are kept in STS conditions at 15 °C and 40–45% RH [90].

### 3.2. Field Genebanks

Although seed desiccation sensitivity affects only about 8% of flowering plants [72], there are many field and horticultural crops as well as (agro)forestry species that cannot be conserved long-term in conventional seed storage and that require different forms of conservation, such as in field genebanks, in in vitro collections, and/or in liquid nitrogen [93]. Among those are species that only produce recalcitrant or intermediate seeds with a short storage life span. Moreover, some species take several years to produce seeds, such as yucca (*Yucca* sp.) and bamboo (a species of the Poaceae subfamily Bambusoideae), while other crop species hardly produce seeds and are only vegetatively propagated, such as edible banana and plantain (*Musa* sp.) [95].

Major food crops that are commonly clonally propagated and therefore conserved in field genebanks include herbs, shrubs, vines, and trees, and these food crops belong to about 34 families [96]. Among those are sub-tropical and tropical shrub and tree species, such as coffee (*Coffea* sp.), cacao (*Theobroma cacao*), rubber (*Hevea brasiliensis*), coconut (*Cocus nucifera*), peach palm (*Bactris gasipaes*), breadfruit (*Artocarpus altilis*), mango (*Mangifera indica*), citrus (*Citrus* sp.), avocado (*Persea americana*) many temperate fruit trees, root and tuber crops such as potato (*Solanum tuberosum*), cassava (*Manihot esculenta*), yams (*Dioscorea* sp.), sweet potato (*Ipomoea batatas*), taro (*Colocasia esculenta*), other aroids, bananas, garlic (*Allium sativum*), shallot (*Allium cepa* var. *aggregatum*), grasses such as sugarcane (*Saccharum officinarum*), and forages. Additionally, temperate and sub-tropical fruit trees like peach (*Prunus persica*) and apricot (*P. armeniaca*) are typically clonally propagated to maintain the genetic constitution of the variety. As their seeds are non-orthodox, i.e., they cannot be dried to low seed moisture content and thus cannot be stored for longer periods at low temperatures, they are maintained in field genebanks and increasingly as in vitro materials (see Section 3.3) or cryopreserved (see Section 3.4). Although some of those crops are sexually fertile, they do not breed true to type, hence, the preferred method is vegetative propagation which enables the maintenance of genotypes as clones.

In field genebanks, the plant genetic resources are kept as live plants that undergo continuous growth and require regular care and maintenance. Accessions maintained in field genebanks need considerable space, especially tree species, and require much more attention in their day-to-day management than seed or in vitro collections, as the plants are continuously exposed to biotic and abiotic stresses. Integrated pest and disease measures are essential to ensure that plants are free of pathogens [97].

Given the exposure of plants in field genebanks to biotic and abiotic stresses and physical security threats (invading animals, theft), these do not present the most secure methods of germplasm conservation; however, they are often the only practical and cost-effective choice to conserve the germplasm of clonal crops, especially when resources and skills for alternative conservation approaches, such as in vitro conservation or cryopreservation, are out of reach.

When field genebank conservation is the only viable alternative, careful planning of site selection and appropriate field management can help to mitigate those risks. The revised and updated Genebank Standards of the FAO [78] indicate the best practices for the safe establishment and management of field genebanks, including the choice of location, the acquisition of germplasm, the establishment of field collections, appropriate field management, the regeneration and propagation of plant material, characterization, evaluation, documentation, distribution, and security and safety duplication.

#### 3.2.1. Risks Associated with Field Genebanks

Adaptation of accessions. If environmental and soil conditions as well as the elevation of the field genebank are quite different from the site where plant material was collected, some poorly adapted accessions may fail to develop properly or may grow much more slowly than better adapted accessions. Moreover, poorly adapted accessions are also more prone to pest and disease infestations, hence losses of individual plants or entire accessions might occur over time. To mitigate such risks, a decentralized field genebank approach might work better, if it is feasible, i.e., the establishment of poorly adapted accessions at sites with agro-ecological conditions that are more like the original collection site [78]. The natural environment of the original site can be simulated to some degree, as is practiced at the international coffee field collection maintained by CATIE in Turrialba, Costa Rica [98]. Dense and almost permanent shade is provided for the wild genotypes from Ethiopia, while the cultivated accessions from East Africa are exposed to full sunshine. Cultivated accessions are grown under light shade, as is the case in commercial coffee production. Curational staff must always pay special attention to the growth and performance of the accessions of wild species to avoid plant losses. Poorly adapted accessions should also be duplicated at alternative sites or grown in greenhouses to avoid the loss of entire accessions. A safe alternative backup option is the cultivation of valuable, irreplaceable accessions in in vitro conditions or their preservation in liquid nitrogen. The latter has been shown to be an interesting long-term conservation approach for coffee germplasm, as cryopreservation costs (in perpetuity per accession) were lower than conservation in field genebanks [99].

Physical safety and plant health considerations. The absence of major threats from natural calamities, such as earthquakes, volcanoes, hurricanes, typhoons, and floods is important when deciding on the location of a field genebank [78]. A safe distance of at least 10 km radius from active volcanoes should be maintained to avoid damage from lava flow and rocks. Areas that are frequently in the path of hurricanes, typhoons, or snow avalanches should be avoided. Firebreaks can be established if bushfires are a known risk. Fencing and security guards will help to avoid vandalism, theft, and damage by large animals. It is good practice to choose a location where the target crop has not been grown previously to avoid the heavy infestation of major pathogenic diseases or insect pests that might cause plant losses or make disease and pest management very costly [97]. Soils might harbor fungal, plasmodiophorid, oomycete, and bacterial pathogens as well as viruses and plant parasitic nematodes, and termites that are detrimental to plant growth and that may lead to plant death. Many of the soil borne diseases are difficult if not impossible to manage and to eradicate with conventional means. The spread of soil-borne fungi (e.g., *Rosellinia* sp.) led to the death of numerous cacao trees and the entire loss of accessions, making it necessary to relocate the international cacao collection conserved by CATIE in Costa Rica to two new alternative sites [100]. Fire blight caused by the bacterium *Erwinia amylovora* is one of the most devastating apple diseases worldwide, and it can severely damage or even eradicate susceptible apple accessions in field genebanks [101]. Given all of the above-mentioned physical safety and plant health challenges with clonally propagated materials, the only safe long-term conservation option for such crops is cryopreservation, which is described further below.

Genetic integrity. Outcrossing species that are used to produce seeds for distribution requires a safe isolation distance to avoid the potential impact of geneflow and contamination from nearby commercial crop stands or from wild populations of the same species [78]. Many forage grasses are out-breeding, and it is recommended to use an isolation distance of at least 100 m between accessions [102]. Larger isolation distances are required for peach palm, as pollination is mainly conducted by insects, particularly small beetles, over distances between 100 and 500 m; wind and gravity can also function as pollen vectors [103]. The maintenance of such large isolation distances is important to preserve rare agronomic traits such as spineless peach palm varieties, e.g., ‘Putumayo’ and ‘tanque de San Carlos’ [104] and make such germplasm with highly sought-after characteristics available for distribution to users. As shown with this specific example, the maintenance of genetic integrity is critical for the facilitation of the use aspect of the PGRFA for direct cultivation or breeding and less so for the long-term conservation of such rare alleles within a population.

Spread of systemic pathogens. While most systemic pathogens are not transmitted via seeds, clonal propagules are often associated with the spread of such pathogens [96]. Therefore, field genebanks as a source of materials for distribution present serious problems for germplasm exchange. Many national or regional disease outbreaks have been associated with the transfer of vegetative propagules, e.g., the spread of banana bunchy top virus (BBTV) to Africa and within the continent, aggressive strains of potato late blight (*Phytophthora infestans*) in Africa and Asia, and potato cyst nematode (*Globodera palladi*) in East Africa, among others [105]. To avoid the spread of dangerous pathogens through the exchange of clonally propagated germplasm, the Germplasm Health Units of the CGIAR recommend the generation of virus-free in vitro plants for germplasm exchange as per the FAO-International Board for Plant Genetic Resources (IBPGR) technical guidelines for the conservation and safe distribution of these crops [106]. All germplasm material exported and imported by CGIAR centers are tested for viruses and other pests as per guidance provided by the National Plant Protection Organizations (NPPO), and only material that is free of viruses and other pests are released to clients. Procedures for germplasm health testing, phytosanitation, and safe international transfers for clonally propagated crops as well as seed crops have been thoroughly reviewed by Kumar et al. [105].

Rejuvenation. Low plant vigour, loss of plants within accessions due to pest and disease pressure, and the high age of plants are major reasons for the rejuvenation of accessions in a field collection. The loss of a single individual plant usually could entail genetic erosion within the accession because there are normally only very few plants representing each accession, sometimes only one individual, especially in the case of woody species. According to Reed et al. [97], the number of replicates is often limited to between 5 and 10 for cassava, 10 and 12 for sweet potato, 1 and 3 for trees and shrubs, 6 and 10 for herbaceous plants, and between 3 and 20 for bananas. In the case of the USDA-ARS apple field collection, for example, trees are grafted in the nursery on M7 dwarfing rootstocks and then planted as duplicates in the fields [107]. Once the primary tree is established, the second tree is removed, thus leaving one grafted tree per accession; hence, there is a clear need to back up a collection to avoid genetic erosion. Regeneration and propagation have species-specific requirements and are very costly management interventions that need to be carefully planned. Rejuvenation might also require relocation to another site to avoid diseases, pests, and soil infestation caused by devastating pathogens. Even handling the entire process of raising rootstocks, vegetative propagation, and replanting to the field with the utmost care, human errors can easily happen, and the accessions can be mixed up [95].

To avoid genetic erosion and the loss of entire accessions, a cryopreservation back-up system is mandatory to safeguard the long-term conservation of important clonal material. Furthermore, safety duplication of field genebank accessions is an essential activity for the security of the conserved genetic diversity.

#### 3.2.2. Advantages of Field Genebanks

Field genebanks provide ready and easy access to the conserved material for characterization, evaluation [108], and research. Phenotypic characterization of accessions in field genebanks is relatively easy to perform, as the plants are readily and permanently available in the field and do not need to be grown out, which is the case for orthodox seed collections. Because of the permanent availability of the plants in the field collection, the scoring of characterization traits can be done at the appropriate time and repeated over the years if necessary [78]. Reference accessions planted in the same field facilitate the correct scoring of specific traits and the interpretation of the results that are obtained. Herbarium specimens and high-quality voucher images will guide true-to-type identification of accessions in a field genebank.

Germplasm users can visit the collections and inspect the plants during the vegetative or reproductive stages to have a first visual impression, which will help in making an informed decision on which germplasm to select and order. Fruits and/or vegetative material are readily available for germplasm distribution. The exposure of vegetatively propagated plants in the field genebank to changing environmental conditions allows for a gradual adaptation process of the plants [96] in contrast to the seeds kept in seed storage in a frozen state over several decades. This may present a major advantage to germplasm users. In combination with the cryopreservation techniques developed for long-term conservation of clonal germplasm, field genebanks facilitate the visual germplasm selection process, while in vitro collections support the safe exchange of clonal plant germplasm.

### 3.3. In Vitro Collections

Alternative conservation strategies for vegetatively propagated crops and species with recalcitrant seeds are in vitro cultures for short- to medium term conservation (MTS) comprising a couple of months up to a few years—the so-called in vitro active genebank (IVAG) [109,110,111]. In the IVAGs, plant material is maintained under slow-growth conditions with species-specific successive subculturing and renewal, readily available for multiplication and distribution to germplasm users. Cryopreservation in liquid nitrogen is the technology available for long-term conservation, denominated in an in vitro base genebank (IVBG). Technical guidelines providing guidance to researchers and genebank and botanic garden managers for the establishment and management of in vitro germplasm collections have been published [95,108], and genebank standards for maintenance of PGRFA in vitro have been developed [78].

Slow-growth culture conditions are applied to in vitro collections to reduce the frequency of subculturing, which is labor-intensive and is a source of contamination of the cultures. Entire accessions might be lost due to handling errors (mixing, mislabeling, misidentification) and genetic instability (somaclonal variation) induced by the tissue culture environment [95]. Under optimal growth conditions, subculture frequencies range from one to three months, whereas under slow growth conditions, the subculture period can vary from one to two years, depending on the crop, the environmental conditions in the culture room, and the media composition. Slow-growth conditions aimed at reducing the metabolic activity of the in vitro plantlets can be achieved by applying physical, chemical, or nutrient growth limitations, either individually or in combination [110,112]. Physical growth limitations are achieved, within limits, by lowering the temperature of the growth room, often in combination with low light intensities and restricted photoperiods. Other measures comprise minimal containment in small culture vessels resulting in conditions that minimize the growth and development of plants by restricting space, gaseous exchange, and nutrient supply [112].

Species from temperate climates are, in general, more cold-tolerant than species from the tropics and subtropics. A low temperature regime of 2 °C and 10 °C is used for the MTS of in vitro grown *Allium* species at the Leibniz Institute of Plant Genetics and Crop Plant Research (IPK), Gatersleben, Germany, extending the culture cycle to 12 months [113]. For MTS storage of potatoes, the IPK applies a multi-step, sequential approach [114]. After the establishment of virus-free potato material, tissue cultures are initially kept at relatively high temperatures of 20 °C under long-day conditions for 2–3 months, followed by a microtuber-induction phase with short-day conditions at 9 °C for 2–4 months and, finally, a cold storage period with microtuber storage at 4 °C for 16–18 months.

Cold-sensitive species from the tropics and subtropics require higher storage temperatures of at least 15 °C for sweet potato [115], 16 °C for *Musa* [116], 21 °C for pineapple [117], and 25 °C for yam, with subculturing intervals of two months [113]. The higher the storage temperature, the shorter the subculture intervals.

A chemical growth limitation involves the application of osmotically active agents such as mannitol, sorbitol, and polyethylene glycol (PEG), resulting in water stress for the tissues or the addition of growth retardants such as abscisic acid (ABA), paclobutrazol, ancymidol, and hydrazides to the culture media [95,112]. A nutritional growth limitation is based on low levels of macro- and micronutrients in the culture medium, resulting in the slow growth of tissue cultures [110].

Combining physical (temperature of 6 °C; 16-h photoperiod), chemical (20 gL^−1^ mannitol inclusion in culture media), and nutritional (40 gL^−1^ sucrose) growth limitations, Sarkar and Naik [118] were able to extend viability of potato microplants in vitro for up to 30 months without subculturing. However, the only long-term conservation option is cryopreservation, which is described in Section 3.4.

#### 3.3.1. Risks Associated with In Vitro Collections

Freedom from contamination. Tissue culture is central to the safe movement of clonal plant germplasm; hence, it is important to assure the purity of the cultures. During the germplasm acquisition process a health test is conducted, and viruses, if present, are eradicated, followed by disease indexing before entering the in vitro genebank [110]. However, it remains possible that covert, systemic endophytes go undetected and continue residing in germplasm tissues after the disease eradication and sterilization procedures. These organisms may become opportunistic pathogens and pandemic agents if spread by vectors such as mites and thrips in the culture room. Furthermore, mites, thrips and other small arthropods may cause the proliferation of fungal contaminations in the tissue cultures, and these are quite difficult to eradicate [97].

Correct identity of accessions. The identity of cultures may be compromised because of human errors, such as the physical mixing of the accession samples and documentation errors due to mislabeling or misidentification [115]. The CGIAR genebanks adopted a rigid authentication process as part of their quality management process that starts with the verification of the documentation that is associated with germplasm acquisition (passport information) followed by testing the incoming accessions with standard markers and descriptors and the application of informatics tools [110]. A wide range of molecular techniques is available to authenticate germplasm [119]. Moreover, DNA barcoding is evolving as a robust technology that allows routine checks for genetic authenticity and ensures that a mistaken identity is not perpetuated [110,120,121]. Electronic barcoding is also an important quality assurance tool that allows instant traceability and provides current information on the status of each accession in the genebank at any point in time. This information needs to be linked to an electronic inventory system to support the retention of authenticated status and to prevent errors arising from transcribing hand-written records.

Somaclonal variation. A problem that is often associated with micro-propagated plants are somaclonal variations, i.e., genetic aberrations that are caused by mutations or epigenetic effects [122]. This is especially the case when tissue is exposed to minimal (slow or sub-optimal) growth conditions over long periods of time and may be due to the accumulation of ethylene, which restricts growth and may exacerbate other stresses induced during slow growth in in vitro storage [110]. In general, the species or crop and the genotype within the same crop, the propagation methods and the nature of the tissue used as the starting material, the type and concentration of growth regulators added to the culture medium as well as the number and the duration of subcultures are some of the factors that determine the frequency of occurrence of somaclonal variation in vitro. Disorganized growth phases in tissue cultures, especially in callus and cell suspension cultures, increase the chances of mutations [122]. Banana is a crop which is frequently affected by somaclonal variation, and with increasing subculture events, the proportion of variants can reach levels of up to 72% [123]. Plant growth regulators present in the culture media seem to indirectly affect somaclonal variation by increasing the multiplication rate of the cultures. To minimize problems with somaclonal variation in micropropagated plants, it is recommended to use organized tissue systems, such as shoot cultures, upon culture initiation, rather than callus and suspension cells, and to culture the plantlets on hormone-free media for medium-term in vitro storage [95].

Cellular ageing and senescence. Cellular ageing leading to a loss of biochemical and physiological functions in cells and senescence are observed during prolonged cultivation in vitro. In eight-year-old peach palm (*Bactris gasipaes*) cultures, initiated through direct morphogenesis of adventitious buds without callus formation, Graner et al. [124] observed generalized senescence and probable ageing in clones.

Safety duplication. To avoid the aforementioned risks, it is mandatory to duplicate the collection, either in vitro or in cryopreservation, and preferably in another distant location to ensure that the duplicate collection is properly secured [95]. For example, the Bioversity International *Musa* Germplasm Transit Centre (ITC) hosted at the Katholieke Universiteit Leuven, Belgium and home to the world’s largest collection of banana diversity, maintains 70% of its in vitro clones in a cryopreserved base collection. A cryopreserved sample of each in vitro clone is safely duplicated at the Institut de Recherche pour le Développement—National Research Institute for Sustainable Development (IRD), Montpellier, France [95].

#### 3.3.2. Advantages of In Vitro Collections

In vitro conservation has several compelling advantages when compared to field genebanks, as accessions are not subjected to the risks of climate variability and pest and disease outbreaks, which are frequent in the latter. The availability of germplasm samples from a field genebank is restricted by the season, and the development stage of the plant and the international movement of vegetative propagules carries inherent risks of transmitting pernicious plant pathogens [95]. In contrast, tissue culture samples are available year-round [112], have a low space requirement, and are characterized by a high multiplication rate [95]. Moreover, tissue cultures are an internationally recognized means of the safe germplasm movement of disease-free material under aseptical conditions [105,106].

By limiting the international movement of vegetatively propagated plants to sterile in vitro plants, intracellular obligate pathogens, such as viruses, viroids, and phytoplasmas, are the only remaining concern [105]. These pathogens can be eliminated through meristem culture, thermotherapy, chemotherapy, electrotherapy, and cryotherapy [110,125,126,127]. In grapevine, electrotherapy consisting of the electric stimulation of grapevine herbaceous cuttings with an electric current of 40–100 mA for 5–20 min in an electrophoresis tank followed by the in vitro regeneration of new plants has been successfully used for the complete and/or partial elimination of viruses [126]. Cryotherapy is an option for pathogen eradication in those crops for which cryopreservation protocols are available, and it has been successfully applied to several crops, such as potato (*Solanum tuberosum*), sweet potato (*Ipomoea batatas*), grapevine (*Vitis vinifera*), citrus (*Citrus* sp.), raspberry (*Rubus idaeus*), banana (*Musa* sp.), apple (*Malus domestica*), kiwifruit (*Actinidia chinensis*), and gentian (*Gentiana triflora*) [127,128]. Detailed protocols for pathogen (virus, viroids) elimination and plant health status verification as they have been applied for banana, cassava (*Manihot esculenta*), potato, sweet potato, and yam (*Dioscorea* sp.) by the CGIAR genebanks have recently been summarized by Kumar et al. [105].

In summary, slow growth in an in vitro culture system is a successful method of securing plant germplasm under medium-term storage conditions, similar to field genebanks. In vitro genebanks that cultivate clonally propagated crops can apply various methods for pathogen elimination, enabling the safe distribution of clonal plant germplasm to users. Apart from field genebanks, it is the only method to conserve crops with recalcitrant seed that cannot (yet) be cryopreserved due to the lack of successful cryopreservation protocols. It is also an essential element for the recovery of cryopreserved plant germplasm and, therefore, an essential link to the long-term conservation of a crop germplasm that does not produce orthodox seed.

### 3.4. Cryopreservation

Given the limitations and problems associated with field genebanks and in vitro collections described above, cryopreservation, i.e., the storage of biological material at an ultra-low temperature, usually in liquid nitrogen (−196 °C) or its vapor phase (between −140 and −180 °C), is the only method that is currently available to ensure the safe and cost-effective long-term conservation of the PGRFA of species that have intermediate or recalcitrant seeds, that hardly produce seeds at all, or that are vegetatively propagated [129]. Cryopreservation can be applied to both in vivo materials, such as seed and dormant buds, as well as to in vitro materials comprising cell suspension and callus cultures, shoot tips, somatic and zygotic embryos, and embryonic axes [130].

Plant cryopreservation studies started about 60 years ago, when Sakai [131] was able to show that cold-hardened tissue sections of mulberry twigs were able to survive exposure to liquid nitrogen when first pre-frozen at −20 °C, a step that led to the dehydration of the freezable water in the cells. He clearly demonstrated that the hardening of the cells through exposure to low winter temperatures and the dehydration of their tissues were essential elements of tissue survival. The formation of ice crystals within the cells of cryopreserved material leads to cell death. Effective dehydration removes all of the freezable water from the cells and leads to the vitrification of the highly concentrated cytoplasm [132]. Vitrification means the transition of water directly from the liquid phase into an amorphous phase or glass, avoiding the formation of lethal ice crystals in the cells [95]. Cryopreservation procedures comprise slow and controlled rate cooling techniques as well as different dehydration techniques prior to direct immersion in liquid nitrogen. The latter include dehydration, vitrification, encapsulation-dehydration, encapsulation-vitrification, pre-growth, pre-growth dehydration, and droplet-vitrification [130,132].

#### 3.4.1. Dehydration

The dehydration of explants intended for cryopreservation is mainly used with seeds, zygotic embryos, or embryonic axes extracted from seeds followed by direct immersion in liquid nitrogen for rapid cooling, except for oily seeds (e.g., *Arachis hypogea*), which require a slow pre-cooling phase in a programmable cooler before cryopreservation [130] and a slow seed imbibition phase over water [133]. The natural cold acclimatization of twigs in combination with dehydration is also a key element for dormant bud cryopreservation, which usually requires controlled-rate cooling [134,135]. At the Millennium Seed Bank of the Royal Botanic Gardens, Kew, desiccation-tolerant, orthodox seeds of wild species with short lifespans under standard long-term conservation conditions (−20 °C) are dried at about 32 ± 3% RH at 18 °C and are then stored in the vapor phase of liquid nitrogen [94]. In general, cryogenic storage extends seed longevity compared to conventional freezer storage [133]. However, the extension of seed longevity seems to be species-specific, and, above all, a high initial seed quality is critical to maximize the benefits of cryostorage [136].

Apart from orthodox seeds, dehydration has also been applied to seeds, embryos, and embryonic axes of a wide range of recalcitrant and intermediate tropical species [137]. Such species are usually dried to a SMC (fresh weight basis) ranging from 10 to 20% [130].

#### 3.4.2. Controlled-Rate Cooling

Controlled-rate cooling is commonly employed for temperate and subtropical species, including dormant buds, apices of cold-tolerant species, and undifferentiated cell cultures, such as callus and cell suspension cultures [130,132], as well as for oily seed species [133]. The use of dormant buds for cryopreservation is a relatively easy and cost-effective cryopreservation method, as it does not involve aseptic cultures and the excision of shoots. An effective protocol for the cryopreservation of dormant apple buds (*Malus* sp.) was developed at the USDA National Center for Genetic Resources Preservation (NCGRP) in Fort Collins Collins, Colorado, USA [134], and more than 2300 apple clones have been cryopreserved and are currently being maintained in liquid nitrogen vapor conditions [107].

Volk et al. [138] provide a detailed description of the apple dormant bud cryopreservation protocol consisting of the following steps: (i) collecting dormant budwood twigs in mid-winter and cutting them into single node segments; (ii) air-dehydrating the twigs at −5 °C and 35% RH to a 25–30% moisture content (fresh weight basis—fwb); (iii) placing the dehydrated twigs in tubes that are heat-sealed, labeled, and placed in cryoboxes for slow freezing in a programmable cooler at 1 °C per hour from −5 °C to −30 °C and holding this temperature for 24 h; (iv) transferring pre-frozen boxes to the vapor phase of liquid nitrogen for long-term storage; (v) allowing the cryopreserved nodal sections to rehydrate at 2–4 °C for 14–21 days on moist peat moss for recovery; and finally, (vi) the rehydrated buds are budded onto potted seedling rootstocks (2 buds per rootstock).

Apart from apples, the described dormant bud cryopreservation has also been successfully developed for pear (*Pyrus* sp.) [139] and sour cherry (*Prunus cerasus*) [140]. Recent studies [141] confirmed that the air drying of dormant budwood to ~30% moisture content followed by slow cooling before liquid nitrogen storage was the most critical pre-storage treatment for increasing freezing resistance and cryosurvival. The fruit crops that were covered in these studies included apple, pear, sweet cherry (*Prunus avium*), apricot (*Prunus armeniaca*), and peach (*Prunus persica*). For peach, the best pre-storage moisture level was slightly higher at 35% (fwb), an indication that desiccation sensitivity may contribute to low cryosurvival. Similar protocols for the cryopreservation of dormant blueberry (*Vaccinium* sp.) are also under development, and it has been shown that the pre-harvest temperature of the twigs (which should remain below 11.2 °C for a 10-day period) is a critical factor for the successful post-cryopreservation viability of blueberry dormant buds [142]. In the case of mulberry (*Morus* sp.) [143] and blackcurrant (*Ribes nigrum*) [144], cryopreserved buds are recovered in vitro before being transferred to the field.

#### 3.4.3. Vitrification-Based Cryopreservation Protocols

Apart from the conventional dehydration of the tissues to be cryopreserved, several protocols make use of the addition of cryoprotectants to increase the viscosity and to achieve suitable cellular dehydration, while avoiding ice formation [145]. A total of seven vitrification-based cryopreservation protocols can be distinguished [129,130], which consist of (i) encapsulation dehydration; (ii) vitrification; (iii) encapsulation-vitrification; (iv) dehydration; (v) pre-growth; (vi) pre-growth dehydration, and (vii) droplet vitrification.

Droplet vitrification is now the most common and most widely used cryopreservation protocol for hydrated tissues, such as in vitro cultures [95]. This method exposes meristem tips to plant vitrification solution (PVS), leading to a more concentrated, vitrifiable cell solution, which can then be exposed to liquid nitrogen for long-term cryostorage [95,132]. Recent modifications to the droplet vitrification method comprise the use of aluminum cryoplates with encapsulation dehydration or encapsulation vitrification [146,147,148]. With these more recent protocols, meristems are enclosed in tiny drops of calcium alginate and placed on the aluminum plate before being dehydrated and subsequently exposed to liquid nitrogen. Cryopreservation by droplet vitrification has been successfully tested in grapevine (*Vitis vinifera*), gentian (*Gentiana triflora*), potato (*Solanum tuberosum*), kiwifruit (*Actinidia chinensis*), and raspberry (*Rubus idaeus*) in New Zealand. This technology is also being applied for the pathogen eradication of viruses and bacteria infecting those crops, thus ensuring the long-term conservation of healthy clonal plant material [127].

Unfortunately, there is no ‘generic cryopreservation protocol’ that can easily be adopted and adapted to a wide range of species. The science and methodology of cryopreservation, i.e., protocol development, is still a major challenge for many crop species. A further difficulty is the successful implementation of available cryopreservation protocols to an entire crop collection, as some genotypes within the same species might not respond favorably to a specific protocol requiring further modifications [95,132,149].

Major cryopreserved collections of temperate, subtropical, and tropical plant species include apple (*Malus* sp.), pear (*Pyrus communis*), *Citrus* sp., mulberry (*Morus* sp.), potato (*Solanum tuberosum*), grape (*Vitis vinifera*), coffee (*Coffea arabica*), *Musa*, sweet potato (*Ipomoea batatas*), cassava (*Manihot esculenta*), yam (*Dioscorea* sp.), strawberries (*Fragaria × ananassa*), hops (*Humulus lupulus*), garlic (*Allium sativum*), chives (*Allium schoenoprasum*), mint (*Mentha* sp.), and medicinal plants (for an overview of conservation institutes, crops conserved, and cryopreservation methods used, please refer to Panis [150]). Recently, O’Brien et al. [148] reported on the successful cryopreservation of the somatic embryos and shoot tips of avocado (*Persea* sp.). It has been estimated that about 100,000 unique accessions of vegetatively propagated and recalcitrant seed crops require long-term conservation through cryopreservation, while currently, about 18,500 accessions are conserved by this method [151], up from the approximately 10,000 accessions reported by Acker et al. [149]. Most cryopreserved accessions belong to five crops: potato (38%), cassava (22%), bananas and plantains (11%), mulberry (12%), and garlic (5%) [149].

#### 3.4.4. Advantages and Limitations of Cryopreservation for Long-Term Conservation

The major benefit of cryopreservation protocols is the fact that this technology is the only available method that allows the safe and long-term conservation of many species that are vegetatively propagated or that have recalcitrant seeds (mostly from the tropics and subtropics), which otherwise can only be conserved in field genebanks or in in vitro collections. The inherent risks and the short- to medium-term nature of these conservation methods have been described above. In general, introducing an accession into cryopreserved storage is more expensive than establishing an accession in in vitro culture or in the field. However, the costs of maintaining an accession in cryopreserved storage for the long-term (above 20 years) are considerably lower than those of maintaining an accession in the field or in vitro, particularly when dealing with a large number of accessions [99,130,132,149]. Moreover, cryopreservation is a conservation method that ensures genetic stability over long periods of time. In addition, cryotherapy offers additional benefits for the removal of viruses from a wide range of vegetatively propagated crops [127,128].

At present, over one million seed samples from national and international genebanks are being conserved at the Svalbard Global Seed Vault (SGSV), Norway, a global security back-up system for long-term seed conservation, at −18 °C [152]. Although clonal crop collections can be duplicated for safety reasons at other locations, either in the field or in vitro, the safest backup approach would be the cryopreservation of a safety duplicate. A recent feasibility study concluded that a safety backup facility similar to the one in Svalbard, Norway is required to accommodate a duplicate of the approximately 10,000 unique accessions currently cryopreserved at the global level and to offer space for additional safety duplicates arising from on-going cryopreservation activities [149]. Unfortunately, the implementation of this important proposal has not yet started.

### 3.5. DNA Banks

DNA storage is regarded as an emerging complementary ex situ technique for safeguarding the genetic diversity of a crop’s genepool, especially for species that are difficult to conserve by conventional means in the form of seeds or vegetatively in field genebanks, in in vitro collections, or via cryopreservation and that are highly threatened in the wild [153]. The transfer of genetic material in the form of DNA samples rather than seed would be especially meaningful for programmes that focus mainly on genetic and genomic studies and not on agronomic performance. It is a lot easier and safer to exchange DNA samples than seed or vegetative propagules, as the latter require seed/planting material inspection, phytosanitary certificates, and post-entry quarantine testing to ensure that the requested genetic plant resources are free from undesirable diseases and pests [154]. Moreover, shipping costs of DNA samples are considerably lower than those of seed or vegetative material.

DNA banks can also serve as backup or safety duplicates of the physical seed, field, or in vitro collections in case of catastrophic losses [154]. Although it is not (yet) possible to recover a plant from a DNA sample, the storage of entire genomes (total DNA) or genome fragments (genomic libraries) would permit the preservation of its valuable genetic information thus contributing to the objective of gene or genome conservation [155,156]. With the impressive advances in molecular genetics, these preserved genes or genomes might be of high relevance in the future. Genome conservation could play a major role for species that are currently under threat of extinction or that are already extinct [156]. While DNA banks are considered as a common genetic biodiversity repository [157], Datlof et al. [158] were able to demonstrate that the tissues of target species stored in DNA banks also harbour their corresponding microbial symbionts, many of which are yet to be discovered.

In anticipation of the emerging role of genomics in the conservation of PGRFA, the International Plant Genetic Resources Institute (IPGRI, now Bioversity International) conducted a global survey on the feasibility of DNA storage and use in 2004 and published its findings in a book on “DNA banks—providing novel options for genebanks” [159]. Guidelines for the management of DNA banks have been reviewed and described by Hodkinson et al. [160]. DNA resources can be maintained at −20 °C for short- and medium-term storage, i.e., up to 2 years, and at −70 °C or in liquid nitrogen for much longer periods, comparable to long-term seed storage [161]. Several factors, such as space, containers, frequency of access, cost, stability (temperature fluctuations), and security (breakdown of equipment) impact decisions about using conventional freezers (−18 to −20 °C), −80 °C freezers, or liquid nitrogen storage [162]. Liquid nitrogen (LN_2_) freezers are the most secure option, as they do not require mechanical compressors; hence, the equipment does not fail in the event of power outages. However, this option is more costly and is primarily used for the long-term storage of hydrated samples. Preservation stresses, such as the drying of tissue to be stored, freezing, or the factor time, may inflict some damage to the DNA, but most chromosomal aberrations are repaired in the surviving cells after a few cell divisions [162].

Purified DNA dissolved in buffer may be safely stored for 1–2 years at 4 °C for 4–7 years at −18 °C and for more than 4 years at −80 °C, however, the overall fragment size and, consequently, the DNA quality decreases with storage time [162]. Long-term DNA conservation can also be achieved using a solid medium, such as cellulose-based cards, instead of DNA dissolved in buffer [154]. The paper conservation method is also an efficient means of inactivating pathogens and protecting plant DNA from degradation. DNA can either be stored within the tissue after transfer to the paper or as extracted DNA after submitting the plant tissue to an extraction protocol and transferring the nucleic acid to the paper. The DNA that is conserved on paper can be safely stored at room temperature and 30% relative humidity, at least for medium-term storage [162]. DNA samples on paper can be easily exchanged among institutions, and identification is facilitated by bar-coded tags that allow for a complete recovery of the sample information.

An interesting further development of the use of paper for DNA storage and exchange is the development of DNA books. DNA clones or PCR products are printed directly onto the pages of books and are delivered to users along with the relevant scientific information [163]. The DNA sheets are not damaged by high temperatures and humidity, conditions that might be imposed on the sheets during bookbinding and delivery to the recipients. Recipients can extract the DNA fragments from the DNA sheets and can amplify them using a polymerase chain reaction (PCR). In this context, we can refer to the Rice Full-Length cDNA Encyclopedia DNABook™, which contains 32,000 clones printed on special paper and is bound as a book [153]. A DNA book allows the efficient maintenance of tens of thousands DNA materials in a small space and under ambient conditions. It is an approach that is much less costly than DNA storage in a freezer and allows distribution using ordinary mail.

However, a study conducted by Colotte et al. [164] clearly demonstrated the necessity of protecting DNA from the air (humidity, ozone) to preserve its integrity at room temperature. Such conditions can be created by DNA encapsulation in laser-sealed capsules and accelerated ageing studies at a high temperature (76 °C) and at 50% RH for 30 h did not show any detectable DNA degradation [165]. Storing DNA samples for longer periods under these accelerated aging conditions required the addition of trehalose, which provides a protective matrix to the encapsulated DNA. By extrapolation, this could correspond to 100 years of storage at 25 °C, according to the Arrhenius model [165]. Therefore, DNA encapsulation seems to be a safe method for long-term DNA storage at room temperature, guaranteeing durable DNA stability and facilitating the international movement of DNA samples for molecular biology research.

Within living organisms, DNA is physically and chemically isolated from the environment, and this barrier can keep DNA intact, sometimes for hundreds of thousands of years, as seen in DNA extracted from frozen environments [166], but it can also protect it from arid, hot environments [167]. Lake sediments have also been suggested for ancient DNA studies [168]. Given the fact that DNA can be degraded during extraction and storage, most DNA banks store cells or tissues and extract DNA upon request [158,162]. Seeds are an efficient and inexpensive means of storing the DNA of individual genotypes. As long as seeds are viable, the supply of DNA is guaranteed. However, even seeds that have lost viability can still be used for DNA extraction and amplification, as has been shown in the case of 70- and 135-year-old seeds that were stored under ambient conditions [169]. This is of special relevance for accessions collected from wild populations, i.e., crop wild relatives, which might be threatened in situ.

As DNA can withstand significant variations in temperature as well as modest variations in moisture and offers tremendous information density, DNA storage is currently being explored beyond biological systems for the safe, long-term preservation of important information, such as a global seed vault [170]. Koch et al. [171] developed a storage architecture, called the DNA of things (DoT), for storing DNA-encoded information in 3D-printed objects. To protect the DNA from degradation at the elevated temperatures of 3D-printing, the DNA is encapsulated in nanometer silica beads and is then fused into the raw materials used for 3D-printing. Through this approach, the molecular memory can be concealed in the object and recovered at any time, even after the object has been damaged [172]. The encoded information can be retrieved by sequencing the DNA that has been extracted from a tiny portion of the object.

### 3.6. Pollen Banks

Pollen conservation is a complementary tool for the management and exchange of plant genetic resources, as it helps to conserve the alleles of a genotype or a population [173]. Pollen storage also facilitates crosses in breeding programmes, for example, for wide crosses, when natural pollen production is low or to overcome flowering asynchrony between parents [174]. Other uses of pollen storage include fertility research and studies in basic physiology, biochemistry, and biotechnology for gene expression, transformation, and in vitro fertilization [175]. Pollen should be harvested at peak anthesis, preferably in the morning hours [176]. To save collecting and processing time, it is often recommended to collect anthers in the field and then to separate the pollen grains from the anthers in the laboratory [173]. Pollen is quite sensitive and deteriorates quickly when kept at room temperature and at high relative humidity (75%) [177].

Cytological studies undertaken with 265 plant families by Brewbaker [178] revealed that about 68% release pollen in the bicellular state at anthesis (e.g., Rosaceae), 20% in the tricellular state (e.g., Compositae), and the remaining 12% releases in both types. The nuclear state of pollen grains at anthesis is a major determining factor for pollen viability during storage. While tricellular pollen has high moisture levels at anthesis (approx. 40–60%) and is desiccation-sensitive, bicellular pollen usually is drier at anthesis and can be safely dried to moisture levels below 10%, and its storage behaviour is similar to that of a desiccation-tolerant seed [175]. Longevity is increased by storing pollen at lower temperatures and at a lower moisture content. Apart from storage conditions (temperature and moisture content), the storage atmosphere can also affect longevity. Freeze-dried and vacuum-dried pollen showed greater longevities when stored in a vacuum compared to storage in air [179]. Similarly, pollen viability was enhanced when stored in nitrogen [180]. The beneficial effects of vacuum and nitrogen atmospheres on pollen viability are especially evident at temperature ranges from −5 °C to ambient conditions [175]. The pollen of several species can be successfully stored at temperatures ranging from 4 °C to −20 °C for the short-term [173].

Long-term storage is required if pollen is intended for germplasm conservation and exchange. In this case, pollen should remain viable and functional for about 10 years [181], and the safest way to achieve this is storage in freezers at −80 °C or in cryogenic storage [174,175,182]. Pollen cryopreservation has been successfully demonstrated for a vast range of horticultural crops as well as for staple food crops, forage grasses, ornamental and medicinal plants, and forestry species [182]. As shown by Ren et al. [183], the longevity of cryopreserved pollen seems to be species-specific. The pollen of 102 ornamental plant species/cultivars affiliated to 32 genera of 14 families showed the following changes in pollen viability after cryogenic storage for about 10 years: 11.7% (12 species/cultivars) had increased viability, 16.7% (17 species/cultivars) had stable viability, and the viability of 71.6% (73 species/cultivars) showed a decreasing trend.

Pollen with high moisture levels does not survive exposure to freezing temperatures, most likely due to intracellular ice formation [175]. Therefore, pollen grains are dehydrated before their immersion in liquid nitrogen using silica gel, saturated salt solutions, or drying in an airflow cabinet or oven at 35 °C [174,184].

Desiccation-sensitive pollen such as maize can also be cryopreserved by partially dehydrating pollen to safe moisture levels where no freezable water exists [185]. The highest maize seed set occurred with pollen grains that were dried to about a 12–20% moisture content. Rapid air-drying using pollen dryers that expose the pollen to air at 20–40% RH and at 20 °C has been shown to be beneficial for desiccation-sensitive species of the Poaceae, extending the tolerance of the pollen to freezing temperatures and their longevity [181].

After cryopreservation, a quick thawing protocol is mostly completed by placing samples in a water bath (37–45 °C) or holding them under running water, as reviewed by Dinato et al. [174]. Dried pollen is susceptible to imbibitional injury during rehydration, and this may significantly reduce their viability [186]. Slow rehydration, which can be achieved by placing the pollen in an environment with high RH for a couple of hours at room temperature, minimizes imbibitional damage to pollen grains [187].

Pollen viability can be assessed through the vital staining of pollen grains with fluorescein diacetate (FDA) or tetrazolium-based stains, through in vitro germination, or through effective in vivo fertilization and subsequent seed production [173].

Despite several limitations, such as low the pollen production of some species, the high labour requirement for collecting pollen, the lack of standardized protocols for pollen processing and viability testing, and difficulties in replenishing pollen supplies when quantities are depleted or when the pollen has deteriorated, pollen remains a valuable genetic resource for long-term conservation in cryogenic storage. Moreover, from a biosecurity point of view, pollen is a relatively safe means of germplasm exchange, as pests and diseases are rarely transferred through pollen [108].

In summary, pollen conservation is an additional tool for the maintenance of plant genetic resources and can assist plant breeders to overcome problems such as flowering asynchrony between different parent genotypes and the production of insufficient pollen in nature. Similar to orthodox seeds, the exchange of pollen is a safe means of germplasm exchange, as harmful pathogens are hardly transferred through pollen. For long-term conservation, pollen needs to be cryopreserved, and protocols have already been established for many species. As in other plant structures, the freezable water content needs to be removed from pollen for cryogenic storage in order to safeguard pollen viability during long-term storage at ultra-low temperatures.

## 4. Need for Complementary Conservation Approaches

The ex situ conservation of crop genetic resources largely takes place in genebanks and, to a lesser extent, in botanic gardens. In the case of wild species, such as the relatives of our crops, they are either conserved in their natural habitat or are collected and stored in genebanks or botanic gardens [188]. A special category of crop genetic resources are primitive varieties and landraces of our crop plants. Many of these are still found on farms as part of traditional production systems, and consequently, such materials are maintained ‘on-farm’ or, when collected for the purpose of PGR conservation, are placed in a genebank. In the case of species that grow in natural habitats but that are used by humans for food or medicine, these are mostly left in nature [189].

The use of in situ and on-farm conservation for the routine conservation of PGRFA had a difficult start and was fiercely debated at the FAO [13]. The strong influence of plant breeders and of those that had food production in mind as the most important objective to counter the strongly increasing genetic erosion in the 1960s resulted in a clear preference for ex situ conservation. However, with the increasing interest to widen the conservation to all cultivated plant species and more difficult crops, such as those producing recalcitrant seed or being vegetatively propagated, have become a target for collecting and conservation.

Driven by the strong push for in situ and on-farm conservation by the CBD during the early 1990s and the realization of the importance to also conserve the ‘difficult crops’, a stronger focus on the use of in situ and on-farm approaches became apparent, which is also true for agricultural crops [13]. This development is based on the fact that in situ conservation allows the conserved materials (typically populations in equilibrium with the ecosystem they occur in or traditional varieties and landraces to be cultivated on farm) to remain part of the natural or agricultural environment, in which evolutionary processes continue to manifest themselves. Thus, adaptation to changing conditions can happen, with or without human intervention. Furthermore, as wild plant species and crops are typically widely growing or being cultivated, respectively, much more genetic diversity within and between species can be conserved. The targeted conservation taxa develop naturally under ‘local conditions’; thus, some of the political and managerial issues that apply to ex situ conservation can be avoided. An additional advantage is that the cost of conservation can be limited, which is predominantly confined to monitoring the genetic and species diversity. In the case of on-farm conservation, a close link between people and crops or species is maintained and allows adaptation to changing environmental, cultural, and economic conditions. The on-farm conservation approach is very suitable for ‘crowdfunded, conducted and orientated’ projects and programmes [190]. Possible disadvantages of this conservation method are the limited access to specific subsets of the resources conserved; the lack of adequate characterization and evaluation of the material; and the potential and continuous danger that farmers abandon the cultivation of traditional landraces because of their frequently disadvantaged competitive status. To conserve a given set of genetic diversity on-farm, it will be required that the traditional agro-ecosystem continues to play a livelihood role for the farmers. Due to the dynamic economic, social, and environmental nature of in situ and on-farm conservation, there is a need for careful monitoring practices [191].

The advantages of ex situ seed conservation are the capability of storing large numbers of accessions in a collection, which is relatively cost-efficient; the reproducibility of the results due to the availability of standardized procedures for all major food crops [78]; the possibility to maintain specific genotypes over time; the ready access of the germplasm for characterization, evaluation, research, and distribution; the perceived secure conservation conditions; the generally better health conditions of conserved material and thus the lower risk of spreading diseases; and possibly more specific aspects [191,192]. It should also be noted that within the ex situ approach, complementarity of specific methods do exist, e.g., maintenance in a field genebank can be complemented by in vitro or even cryopreservation storage, as mentioned in the previous section.

The drawback of ex situ conservation is that the germplasm materials are under static and artificial genebank conditions during storage; thus, these accessions are ‘only’ exposed to the selection pressures that are caused by these artificial environmental conditions and not by the (dynamic) natural environmental or cultivation conditions under which the conserved materials could evolve and adapt to the changing conditions.

To facilitate decision-making regarding which conservation method(s) to apply, it is important to know the strengths and weaknesses of both in situ and ex situ methods. The reproductive biology of the species is certainly the most critical one [193]. Genetic erosion and other threat considerations will certainly impact the urgency and the coverage of the genetic diversity that one must address through conservation efforts. Furthermore, it is also important to realize that some of the decision criteria will depend on other factors, such as available infrastructure, trained staff, budget, and the prevailing legal and policy framework as well as collaboration with other institutions inside and outside the country. Furthermore, when planning complementary conservation strategies, the following additional points might also be relevant to consider: the extent of the gene pool coverage and the distribution of the genetic diversity, both within the gene pool as well as geographically [7]. The reproductive biology of a species is critically important to decide which methods are applicable. The extent of genetic erosion and other threats need to be considered [194] as well as non-biological aspects, including the socio-economic feasibility, possible support from governmental agencies, and the availability of markets (in the case of on-farm conservation of traditional crops) are other aspects to take into consideration when deciding on the combination of available conservation methods [191,195]. At the end of the day, it will have to be practical, long-term, and sustainable aspects that should prevail.

The CBD explicitly states that in situ conservation should be given the highest priority but also states that ex situ conservation has an important role to play. Considering the pros and cons of the various conservation approaches, the prevailing conclusions and recommendations are that in situ and ex situ conservation should be combined to achieve more sustainability, long-term security, efficiency, and cost-effectiveness of PGRFA conservation [191,192,196]. Specifically for the efficient protection of crop wild relatives, the concept of so-called trans situ conservation has been introduced, which dynamically integrates multiple in situ and ex situ measures, from conservation to research to education, spanning local to global scales [197]. The conservation of wild chili (*Capsicum annuum* L. var. *glabriusculum*) in southern Arizona is demonstrating this evolving concept.

## 5. Concluding Remarks

The history of the creation and growth of the global conservation system, particularly of the international network of ex situ collections, provides a useful foundation for the critical review of this global system. This foundation is further strengthened by a detailed analysis of the routine genebank operations and of the importance to aim for an integration of in situ and ex situ conservation approaches. In part two of this paper, we will critically review routine germplasm conservation activities, including the active and base collection concept, evaluate new developments that facilitate germplasm conservation and use, assess factors that facilitate or limit the participation of genebanks in the global system, and provide a concluding long-term perspective for an efficient and effective global conservation system.

## Figures and Tables

**Table 1 plants-10-01557-t001:** Historical events of relevance to the establishment and evolution of the global PGRFA conservation, including the international network of base collections.

Year	Event	Main Outputs and (References)	Underpinning Principles (Reference)
Since 1920	Establishment of first genebanks	VIR, St. Petersburg (1920); Commonwealth Potato Collection, Cambridge (<2nd World War); research collections by Rockefeller Foundation, USA (1943); Fort Collins, CO, USA (1958) [12]	Recognition of genetic erosion in landraces by [14]
1926	Publication *Studies on the Origin* *of Cultivated Plants* by N. Vavilov	Monograph in *Bulletin of Applied Botany and Plant-Breeding*; [11]	‘This monograph, dedicated to the memory of De Candolle, seems to be the most substantial contribution made since his day to the history of our main cultivated plants’ [23].
1960	Founding of IRRI	Jointly established by Government of the Philippines’ and the Ford and Rockefeller Foundations [24]	One of the first international genebanks; focus on rice genepool.
1961	Technical Meeting on Plant Exploration and Introduction, FAO Rome	Report of the meeting [15]	Mission-driven approach: conservation and use closely linked, tied to plant breeding, dominance of ex situ collections, mainly in developed countries.
1965	Establishment of the FAO Panel of Experts on Plant Exploration and Introduction.	Six meetings and reports of same during period from 1967–1975 [16]	Formulation of criteria, standards, and procedures for the conservation and use of PGR.
1966	Formal establishment of CIMMYT	Joint Mexican—Ford Foundation breeding project in progress since 1943 [25]	Norman Borlaug awarded Nobel Peace Prize (as wheat breeder) in 1970.
1966	EUCARPIA meeting	Recommendation to foster continental collaboration through the establishment of four sub-regional genebanks in Europe [12]	First indications of establishing a (global) conservation system or network.
1967	FAO/IBP (first) Technical Conference on Plant Exploration, Utilization and Conservation of Plant Genetic Resources, Rome	Publication of *Genetic Resources in Plants—Their Exploration and Conservation* [18]	Need for surveys; concern about genetic erosion of landraces and wild relatives; long-term ex situ collections; guidelines for establishment of global network for ex situ long-term conservation; international collaboration; in situ conservation as a complementary strategy.
1969	Third Session of the FAO Panel of Experts on Plant Exploration and Introduction, Rome	Report [3]	Establishment of collecting priorities by crops (and later) by regions.
1971	Second FAO Technical Conference on crop genetic resources, Rome, Italy	Book on *Crop Genetic Resources for Today and Tomorrow* [19]	Plan of action agreed; panel of experts formulated basic criteria for conservation and use of genetic material (availability; maintaining genetic variability for the long-term; categorizing ex situ collections: base, active, and working collections.
1973	FAO/IBP Technical Conference on Genetic Resources, Rome, Italy	Plan of Action [19]	Recommendation to establish in situ collections.
1974	Establishment of IBPGR	Established as secretariat for its board of trustees, administered by FAO and, technically, as one of the international centres of the CGIAR [26]	Expected to coordinate global exploration and collecting efforts and to orchestrate a global network of genebanks.
1981	Third FAO, UNEP and IBPGR Technical Conference on PGR, Rome, Italy	Report [21]	Clear focus on routine genebank operations; in vitro and in situ (CWRs) conservation; concerns about NUS.
1983	22nd Session of the FAO Conference, Rome, Italy	Adoption of the International Undertaking on Plant Genetic Resources; establishment of the Commission on Plant Genetic Resources for Food and Agriculture (CGRFA) and of the Global System on Plant Genetic Resources [27]	Shared principles; IU non-legally binding; PGRs are a common heritage of humankind; genetic stocks and breeding lines included; germplasm exchange through a network of genebanks; commission provides oversight to system.
1989	3rd Regular Session of Commission on GRFA, Rome, Italy	Call for the development of the International Network of Ex Situ Collections under the Auspices of FAO [28]	Lack of clarity regarding the legal situation of the ex situ collections.
1989	25th Session of the FAO Conference, Rome, Italy	Resolution 4/89: Adoption of an agreed interpretation of the IU; Resolution 5/89: Farmers’ Rights [29]	Plant breeders’ rights are not inconsistent with IU; recognition of Farmers’ Rights.
1991	26th Session of the FAO Conference, Rome, Italy	Resolution 3/91 [30]	Recognition of the sovereign rights of nations over their PGRFA; agreement on development of 1st State of the World’s PGRFA and Global Plan of Action on PGR.
1992	UN Conference on Environment and Development (UNCED), Rio de Janeiro, Brazil	Convention on Biological Diversity (CBD) (entered into force on 29 December 1993);	Biodiversity vs. genetic resources; national sovereignty of states over their resources.
		Chapter 14 of Agenda 21	Call for the strengthening of the FAO Global System on Plant Genetic Resources.
		Chapter 16 of Agenda 21	Biotechnology can assist in the conservation of biological resources (e.g., ex situ techniques); risk assessment of LMOs, biosafety issues.
		Adoption of Resolution 3 of the Nairobi Final Act [31]	Recognises matters not addressed by the convention: a. access to existing ex situ collections; b. questions on Farmers’ Rights; requests FAO forum to address these matters.
1994	1st Extraordinary Session of the CGRFA, Rome	Start of negotiations for revision of IU; 12 centres of CGIAR sign agreement with FAO, placing their collections under the Auspices of FAO [32])	CGIAR centres agree to hold the designated germplasm in trust for the benefit of the international community.
1996	4th International Technical Conference on PGR, Leipzig, Germany	Global Plan of Action for the Conservation and Sustainable Use of PGRFA [21]; First Report on the State of the World’s PGRFA [33]	Recognition of in situ and ex situ approaches; fair and equitable sharing of benefits arising from the use of PGRFA.
2001	31st Session of the FAO Conference, Rome, Italy	Resolution 3/2001: adoption of the International Treaty (entered into force on 11 September 2004) [34]	A legally binding agreement; recognition of Farmers’ Rights (a national responsibility); access and benefit-sharing
2004	Establishment of the Global Crop Diversity Trust	Endowment fund, the income from which will be used to support the conservation of distinct and important crop diversity in perpetuity through existing institutions [35].	Coordinates the Genebank Platform (of the CGIAR operated genebanks)
2006	First meeting of the Governing Body of the ITPGRFA, Madrid, Spain	Standard Material Transfer Agreement (SMTA); relationship between the Treaty and the Crop Trust; agreement between GB and CGIAR centres (Art. 15) [36].	SMTA is the legal instrument through which the MLS operates; recognition of the Crop Trust as an ‘essential element’ of the Treaty’s funding strategy; ex situ genebank collections of CGIAR are put under the Treaty (replacing agreement between CG centres and FAO).
2008	Establishment of the Svalbard Global Seed Vault	Agreement [37].	Additional safety back-up for long-term ex situ collections.
2009	12th Regular Session of the CGRFA, Rome, Italy	Second Report on the State of the World’s PGRFA [38]	Report developed through a participatory approach with member countries
2011	143rd Session of the FAO Council, Rome, Italy	Second Global Plan of Action for the Conservation and Sustainable Use of PGRFA [39]	Need for a roadmap on climate change and genetic resources for food and agriculture

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
