# Peer review of "A Critical Review of the Current Global Ex Situ Conservation System for Plant Agrobiodiversity. I. History of the Development of the Global System in the Context of the Political/Legal Framework and Its Major Conservation Components"

_plants, 2021, doi:10.3390/plants10081557_

Round 1

Reviewer 1 Report

An excellent review and historical synopsis of the development of global ex situ networks. This paper is very well organized and nicely presented I believe it should be published after the minor comments below are addressed:

Line 50: need a citation about the age of plant breeding. 

Line 81: "breeding and other research materials" does not fit in a list of PGRFA. 

Throughout: please do not use plurals like "1960ies", instead use "1960s"

Author Response

Dear Editor,

This is with reference to the comments received from Reviewer 1 on our paper 'A Critical Review of the Current Global Ex Situ Conservation System for Plant Agrobiodiversity. I. History of the Development of the Global System in the Context of the Political/Legal Framework and its Major Conservation Components'. We do appreciate the review and would like to provide our point by point  comments on the review report.

  • We highly appreciate the very nice observation on the draft paper we submitted to Plants as a paper for the Special Issue and thank the reviewer for this.
  • Line 50: need a citation about the age of plant breeding. A precise date on the precise date of plant breeding is difficult to give. Schlegel 2021 has been added.
  • Line 81: "breeding and other research materials" does not fit in a list of PGRFA. This is a point of debate. We have adjusted the statement to also keep the option that these materials are part of the PGRFA open.

  • Throughout: please do not use plurals like "1960ies", instead use "1960s". This suggestion has been corrected throughout the text.

With thanks and kind regards,

Johannes Engels  and Andreas Ebert

Reviewer 2 Report

I believe that this broad review is relatively well written. From a formal point of view, he has nothing to complain about.
In terms of vocational content, I believe that such a text oriented in this way must contain relevant information about:
1) ECPGR - The European Cooperative Program for Plant Genetic Resources (ECPGR).
2) Furthermore, important fruit species such as apricot and peach are being ignored, and cryopreservation is also being worked on - among other things, what I miss in the text.
In general, these species are ignored throughout the work and need to be supplemented - therefore, if it is a full-fledged review.

Author Response

Dear Editor,

We are thankful for the comments of Reviewer 2 and appreciate the general comment on the paper as well as the suggested omissions. These have been added to the text.

  • Specifically, we have added details on the establishment of ECPGR in Section 2.2.
  • With respect to temperate fruit trees, these have indeed not been specifically addressed. A paragraph has been added to Sections 3.2 and 3.4.2. to correct this omission. For details on the conservation methodologies that can be used, reference is made to the corresponding subsections on ex situ conservation methodologies.

With thanks and kind regards,

Johannes Engels and Andreas Ebert
